# Measuring and Controlling Bias for Some Bayesian Inferences and the Relation to Frequentist Criteria

**DOI:** 10.3390/e23020190

**Published:** 2021-02-04

**Authors:** Michael Evans, Yang Guo

**Affiliations:** Department of Statistical Sciences, University of Toronto, Toronto, ON M5G 1Z5, Canada; ygj.guo@mail.utoronto.ca

**Keywords:** principle of evidence, bias against, bias in favor, plausible region, frequentism, confidence

## Abstract

A common concern with Bayesian methodology in scientific contexts is that inferences can be heavily influenced by subjective biases. As presented here, there are two types of bias for some quantity of interest: bias against and bias in favor. Based upon the principle of evidence, it is shown how to measure and control these biases for both hypothesis assessment and estimation problems. Optimality results are established for the principle of evidence as the basis of the approach to these problems. A close relationship is established between measuring bias in Bayesian inferences and frequentist properties that hold for any proper prior. This leads to a possible resolution to an apparent conflict between these approaches to statistical reasoning. Frequentism is seen as establishing figures of merit for a statistical study, while Bayes determines the inferences based upon statistical evidence.

## 1. Introduction

A serious concern with Bayesian methodology is that the choice of the prior could result in conclusions that to some degree are predetermined before seeing the data. In certain circumstances, this is correct. This can be seen by considering the problem associated with what is known as the Jeffreys–Lindley paradox where posterior probabilities of hypotheses, as well as associated Bayes factors, will produce increasing support for the hypothesis as the prior becomes more diffuse. Thus, while one may feel that a very diffuse prior is putting in very little information, it is in fact biasing the results in favor of the hypothesis in the sense that that there is a significant prior probability that evidence will be found in favor of the hypothesized value when it is false. It has been argued, see [1,2], that the measurement and control of bias is a key element of a Bayesian analysis as, without it, and the assurance that bias is minimal, the validity of any inference is suspect.

While attempts have been made to avoid the Jeffreys–Lindley paradox through the choice of the prior, modifying the prior to avoid bias is contrary to the ideals of a Bayesian analysis which requires the elicitation of a prior based upon knowledge of the phenomenon under study. Why should one change such a prior because of bias? Indeed, there is bias in favor and bias against and typically choosing a prior to minimize one type of bias simply increases the other. Roughly speaking, in a hypothesis assessment problem, bias against means that there is a significant prior probability of finding evidence against a hypothesized value when it is true, and bias in favor means that there is a significant prior probability of finding evidence in favor of a hypothesized value when it is false. The real method for controlling bias of both types is through the amount of data collected. Bias can be measured post-hoc, and it then provides a way to assess the weight that should be given to the results of an analysis. For example, if a study concludes that there is evidence in favor of a hypothesis, but it can be shown that there was a high prior probability that such evidence would be obtained, then the results of such an analysis can’t be considered to be reliable.

Previous discussion concerning bias for Bayesian methodology has focused on hypothesis assessment and, in many ways, this is a natural starting point. This paper is concerned with adding some aspects to those developments and to extending the approach to estimation and prediction problems as discussed in Section 3.3 where bias in favor and bias against are expressed in terms of a priori coverage probabilities. Furthermore, it is argued that measuring and controlling bias is essentially frequentist in nature. Although not the same, it is convenient to think of bias against in a hypothesis assessment problem as playing a role similar to the size in a frequentist hypothesis test or, in an estimation problem, playing a role similar to 1 minus the coverage probability of a confidence region. Bias in favor can be thought of as somewhat similar to power in a hypothesis assessment problem and simlar to the probability of a confidence region covering a false value in an estimation problem. Thus, consideration of bias leads to a degree of unification between different ways of thinking about statistical reasoning.

The measurement of bias, and thus its control, is dependent upon measuring evidence. The *principle of evidence* is adopted here: *evidence in favor of a specific value of an unknown occurs when the posterior probability of the value is greater than its prior probability, evidence against occurs when the posterior probability of the value is less than its prior probability and there is no evidence either way when these are equal.* The major part of what is discussed here depends only on this simple principle, but sometimes a numerical measure is needed and, for this, we use the *relative belief ratio* defined as the ratio of the posterior to prior probability. The relative belief ratio is related to the Bayes factor but has some nicer properties such as providing a measure of the evidence for each value of a parameter without the need to modify the prior.

The inferences discussed here are based on the relative belief ratio and these inferences are invariant to any 1–1, increasing function of this quantity. For example, the logarithm of the relative belief ratio can be used instead to derive inferences. The expected value of the logarithm of the relative belief ratio under the posterior is the relative entropy, also called the Kullback–Leibler divergence, between the posterior and prior. This is an object of considerable interest in and of itself and, from the perspective of measuring evidence, can be considered as a measure of how much evidence the observed data are providing about the unknown parameter value in question. This aspect does not play a role here, however, but indicates a close association between the measurement of statistical evidence and the concept of entropy. In addition, many divergence measures involve the relative belief ratio and play a role in [3], which is concerned with checking for prior-data conflict.

There is not much discussion in the Bayesian literature of the notion of bias in the sense that is meant here. There is considerable discussion, however, concerning the Jeffreys–Lindley paradox and our position is that bias plays a key role in the issues that arise. Relevant recent papers on this include [4,5,6,7,8,9], and these contain extensive background references. Ref. [10] is concerned with the validation of quantum theory using Bayesian methodology applied to well-known data sets, and the principle of evidence and an assessment of the bias play a key role in the argument.

As already noted, the approach to inference and the measurement of bias adopted here is dependent on the principle of evidence. This principle is not well-known in the statistical community and so Section 2 contains a discussion of this principle and why it is felt to be an appropriate basis for the development of a theory of inference. In Section 3, the concepts that underlie our approach to bias measurement are defined, and their properties are considered and illustrated via a simple example where the Jeffreys–Lindley paradox is relevant. In addition, it is seen that a well-known *p*-value does not satisfy the principle of evidence but can still be used to characterize evidence for or against provided the significance level goes to 0 with increasing sample size or increasing diffuseness of the prior. In Section 4, the relationship with frequentism is discussed and a number of optimality results are established for the approach taken here to measure and control bias. In Section 5, a variety of examples are considered and analyzed from the point-of-view of bias. All proofs of theorems are in the Appendix A.

## 2. Statistical Evidence

Attempts to develop a theory of inference based upon a definition, or at least provide a characterization, of statistical evidence exist in the statistical literature. For example, see [2,11,12,13,14,15,16,17]. The treatments in [12,14] have some aspects in common with the approach taken here, but there are also substantial differences. There is a significant amount of discussion of statistical evidence in the philosophy of science literature and this is much closer in spirit to the treatment here. For example, see [18] p. 6, where it is stated “for a fact *e* to be evidence that a hypothesis *h* is true, it is both necessary and sufficient for *e* to increase *h*’s probability over its prior probability” which is what is called the principle of evidence here.

### 2.1. The Principle of Evidence

One characteristic of our, and the philosophical, treatment is that evidence is a probabilistic concept and thus a proper definition only requires a single probability model as opposed to a statistical model. This explains in part why our treatment requires a proper prior as then there is a joint probability model for the model parameter and data. The following two examples illustrate the relevance of characterizing evidence in such a context. Example 1 is a simple game of chance where the probabilities in question are unambiguous. The utility aspects of the game are ignored because these are irrelevant to the discussion of evidence but surely are relevant if some action like betting was involved. This is characteristic of the treatment here where loss functions play no role in the characterization of evidence but do play a role in determining actions when required as discussed in the well-known Example 2. The examples also illustrate that characterizing evidence in favor of or against is not enough, as it is necessary to also say something about the strength of the evidence.

**Example** **1.**
*Card game.*


Suppose that there are two players in a card game, labeled I and II, and each is dealt *m* cards, where 2≤m≤26, from a randomly shuffled deck of 52 playing cards. Further suppose that player I, after seeing their hand, is concerned, for whatever reason dependent on the rules of the game, with the truth or falsity of the hypothesis $H_{0}$: player II has exactly two aces. It seems clear that the hand of player I will contain evidence concerning this. For example, if player I has three or four aces in their hand, then there is categorical evidence that $H_{0}$ is false. However, what about the evidence when the event observed is $C_{k}=$ “the number of aces in the hand of player I is k” with k=0,1, or 2?

There are two questions to be answered: (i) is there evidence in favor of or against $H_{0}$ and (ii) how strong is this evidence? The prior probability $P(H_{0})$ and posterior probability $P(H_{0}\;|\;C_{k})$ that $H_{0}$ is true are provided in Table 1 for various (k,m). What conclusions can be drawn from this table? In every case, other than (m,k)=(25,2),(26,2), the conditional probability $P(H_{0}\;|\;C_{k})$ does not support $H_{0}$ being true. In fact, in many cases, some would argue that the value of this probability indicates evidence against $H_{0}.$ This points to a significant problem with trying to use probabilities to determine evidence, as it is not at all clear what the cutoff should be to determine evidence for or against $H_{0}.$ It seems clear, however, that if the data, here the observation that $C_{k}$ is true, has increased belief in $H_{0}$ over initial beliefs, then there is evidence in the data pointing to the truth of $H_{0}.$ Whether or not the posterior probability is greater than the prior probability is indicated by $RB\left(H_{0}\;|\;C_{k}\right)=P\left(H_{0}\;|\;C_{k}\right)/P\left(H_{0}\;\right)$, the relative belief ratio of $H_{0},$ being greater than 1. Certainly, it is intuitive that, when k=0, then our belief in $H_{0}$ being true, a posteriori could increase, but, from the table and some reflection, it is clear that this cannot always be true as the amount of data, *m* in this case, grows. While k=0 is evidence in favor of $H_{0}$, it is evidence against only for m=25,26. The relationship between the prior probabilities and posterior probabilities is somewhat subtle and not easy to predict, but a comparison of these quantities makes it clear when there is evidence in favor of $H_{0}$ and when there isn’t. This answers question (i).

The measurement of the strength of evidence is not always obvious, but, in this case, effectively a binary event, the posterior probability of the event in question seems like a reasonable approach as it is measuring the belief that the event in question is true. Thus, if we get evidence in favor of $H_{0}$ and $P(H_{0}\;|\;C_{k})$ is small, then this suggests that the evidence can only be regarded as weak and similarly if there is evidence against $H_{0}$ and $P(H_{0}\;|\;C_{k})$ is large, then there is only weak evidence against $H_{0}$. Some might argue that a large value of $P(H_{0}\;|\;C_{k})$ should always be evidence in favor of $H_{0}$, but note that the data could contradict this by resulting in a decrease from a larger initial probability. Measuring strength in this way, the table indicates that there is strong evidence in favor of $H_{0}$ with (m,k)=(25,2),(26,2) and weak to moderate evidence in favor otherwise when $RB(H_{0}\;|\;C_{k})>1.$ By contrast, there is typically quite strong evidence against $H_{0}$ in cases where $RB(H_{0}\;|\;C_{k})<1$ with the exception of (m,k)=(10,1),(25,1). Intuitively, it couldn’t be expected that there would be strong evidence in favor of $H_{0}$ for small *m*, but there can still be evidence in favor. Note that a comparison, for m=2and 20, of the values of $RB(H_{0}\;|\;C_{0})$ illustrates that the relative belief ratio itself does not provide a measure of the strength of the evidence in favor. In general, the value of a relative belief ratio needs to be calibrated and the posterior probability of $H_{0}$ is a natural way to do this here.

**Example** **2.**
*Prosecutor’s fallacy.*


Assume a uniform probability distribution on a population of size *N* of which some member has committed a crime. DNA evidence has been left at the crime scene and suppose that this trait is shared by m≪N of the population. A prosecutor is criticized because they conclude that, because the trait is rare and a particular member possesses the trait, they are guilty. In fact, P(“has trait”|“guilty”) =1 is misinterpreted as the probability of guilt rather than P(“guilty”|“has trait”) =1/m, which is small if *m* is large. However, this probability does not reflect the evidence of guilt. If you have the trait, then clearly this is evidence in favor of guilt and indeed RB(“guilty”|“has trait”) =N/m>1 and P(“guilty”|“has trait”) =1/m. Thus, there is evidence of guilt, and the prosecutor is correct to conclude this. However, the evidence is weak whenever *m* is large and a conviction then does not seem appropriate. Since the posterior probability of “not guilty”is large whenever *m* is, it may seem obvious to conclude this. However, suppose that “guilty” corresponds to being a carrier of a highly infectious deadly disease and “has trait” corresponds to some positive, but not definitive, test for this. The same numbers should undoubtedly lead to a quarantine. Thus, the utilities determine the action taken and not just the evidence.

### 2.2. Confirmation Theory

As noted, discussion concerning statistical evidence has a long history, although mainly in the philosophy of science literature, where it is sometimes referred to as confirmation theory. An introduction to confirmation theory can be found in [19], but the history of this topic is much older. For example, see Appendix ix in [20] where, with *x* and *y* denoting events, the following is stated.

If we are asked to give a criterion of the fact that the evidence *y* supports or corroborates a statement x, the most obvious reply is: that *y*
*increases* the probability of x.

The book [20] references older papers and some sources cite [21] where the relative belief ratio RB(A|B) is called the *coefficient of influence of B upon A*. In the Confirmation entry in [22], the definition of *probabilistic relevance confirmation* is what has been called here the principle of evidence. The following quote is from the third paragraph of this entry and it underlines the importance of this topic.

Confirmation theory has proven a rather difficult endeavour. In principle, it would aim at providing understanding and guidance for tasks such as diagnosis, prediction, and learning in virtually any area of inquiry. However, popular accounts of confirmation have often been taken to run into trouble even when faced with toy philosophical examples. Be that as it may, there is at least one real-world kind of activity that has remained a prevalent target and benchmark, i.e., scientific reasoning, and especially key episodes from the history of modern and contemporary natural science. The motivation for this is easily figured out. Mature sciences seem to have been uniquely effective in relying on observed evidence to establish extremely general, powerful, and sophisticated theories. Indeed, being capable of receiving genuine support from empirical evidence is itself a very distinctive trait of scientific hypotheses as compared to other kinds of statements. A philosophical characterization of what science is would then seem to require an understanding of the logic of confirmation. In addition, thus, traditionally, confirmation theory has come to be a central concern of philosophers of science.

As far as we know, Ref. [2] summarizes one of the first attempts to use the principle of evidence as a basis for a theory of statistical inference. Some of the paradoxes/puzzles that arise in the philosophical literature, such as Hempel’s the Raven paradox, are discussed there. Adding the measurement of the strength of evidence and the a priori measurement of bias to the principle of evidence leads to the resolution of many difficulties, see [2]. Whether one is convinced of the value of the principle of evidence or not, this is an idea that needs to be better known and investigated by statisticians.

### 2.3. Popper’s Principle of Science as Falsification

Another aspect requiring comment is that the principle of evidence allows for finding either evidence against or evidence in favor of a hypothesis while, for example, a *p*-value cannot find evidence in favor. This one-sided aspect of a *p*-value is often justified by Popper’s idea that the role of science lies in falsification of hypotheses and not their confirmation. In the context of Examples 1 and 2, this seems wrong as the hypothesis in question is either true or false, so it is desirable to be able to find evidence either way. When applied to a statistical context, at least as formulated in Section 3, inferences about a quantity of interest are dependent on the choice of a statistical model and a prior. It is well understood that the model is typically false and it isn’t meaningful to talk of the truth or falsity of the prior. Since there is only one chosen model, it can only be falsified via model checking rather than confirmed, namely, determining if the observed data are in the tails of every distribution in the model. Actually, all that is being asked in such a procedure is whether or not the model is at least reasonably compatible with the observed data. Similarly, the prior is checked through checking for prior-data conflict, namely, given that the model has passed its check, is there an indication that the true value lies in the tails of the prior. For example, see [3,23,24] for some discussion. Again, all that is being asked is whether or not the prior is at least reasonably compatible with the data.

For checking the model or checking the prior, there is one object that is being considered. Thus, it makes sense that only an indication that the entity in question is not appropriate is available, and a *p*-value can play a role in this aspect of a statistical argument. However, when making an inference, the model is accepted as being correct and, as such, one of the distributions in the model is true, and so it is natural to want to be able to find evidence in favor of or against a specific value of an object dependent on the true distribution. This situation is analogous to what arises in logic where a sound argument is distinguished from a valid argument. A logical argument is based upon premises and rules of inference like modus ponens. An argument is valid if the rules of logic are correctly applied to obtain the conclusions. However, an argument is sound only if the argument is valid and the premises are true. It is a basic rule of logical reasoning that one doesn’t confound the correctness of the argument with the correctness of the premises. In the statistical context, there may indeed be problems with the model or prior, but the inference step, which assumes the correctness of the model and prior, needs to be able to find evidence in favor as well as evidence against a particular value of the object of interest. As part of the general approach as presented in [2], both model checking and checking for prior-data conflict are advocated before inference. If there are serious problems with either, then modifications of the ingredients are in order, but this is not the topic of this paper where it is assumed that the model and prior are acceptable. Thus, Popper’s falsification idea plays a role but not in the inference step.

## 3. Evidence and Bias

For the discussion here, there is a model $\{f_{\theta }:\theta \in \Theta \},$ given by densities $f_{\theta },$ for data *x* and a proper prior probability distribution given by density π. It is supposed that interest is in inferences about ψ=Ψ(θ), where Ψ:Θ→Ψ is onto and for economy the same notation is used for the function and its range. For the most part, it is safe to assume all the probability distributions are discrete with results for the continuous case obtained by taking limits.

A measure of the evidence that ψ∈Ψ is the true value is given by the relative belief ratio
(1)$$RB_{\Psi }\left(\psi \;|\;x\right)=\lim_{\delta \to 0}\Pi_{\Psi }\left(N_{\delta }\left(\psi \right)\;|\;x\right)/\Pi_{\Psi }\left(N_{\delta }\left(\psi \right)\right)=\pi_{\Psi }\left(\psi \;|\;x\right)/\pi_{\Psi }\left(\psi \right)$$
where $\Pi_{\Psi },\Pi_{\Psi }\left(\cdot \;|\;x\right)$ are the prior and posterior probability measures of Ψ with densities $\pi_{\Psi }$ and $\pi_{\Psi }\left(\cdot \;|\;x\right),$ respectively, and $N_{\delta }\left(\psi \right)$ is a sequence of sets converging nicely to {ψ}. The last equality in (Equation 1) requires some conditions, but the prior density positive and continuous at ψ is enough. In addition, when $\Psi =I_{A}$ for A⊂Θ, the indicator of *A*, then we write RB(A|x) for $RB_{\Psi }\left(1\;|\;x\right).$ Thus, $RB_{\Psi }\left(\psi \;|\;x\right)>1$ implies evidence for the true value being ψ, etc. It is also possible that a prior is dependent on previous data. In such a situation, it is natural to replace $\pi_{\Psi }$ in (Equation 1) by the initial prior, as the posterior remains the same, but now the evidence measure is based on all of the observed data. There may be contexts, however, where the concern is only with the evidence provided by the additional data, for example, as when new data arise from random sampling from the relevant population(s), but the first dataset came from an observational study.

Any *valid* measure of evidence should satisfy the principle of evidence, namely, the existence of a cut-off value that determines evidence for or against as prescribed by the principle. Naturally, this cut-off is 1 for the relative belief ratio. The Bayes factor is also a valid measure of evidence and with the same cut-off. When $\Pi_{\Psi }\left(A\right)>0$, then the Bayes factor of *A* equals $RB\left(A\;|\;x\right)/RB\left(A^{c}\;|\;x\right)$ and thus can be defined in terms of the relative belief ratio, but not conversely. In addition, RB(A|x)>1 iff $RB(A^{c}\;|\;x)<1$ and thus the Bayes factor is not really a comparison of the evidence for *A* being true with the evidence for its negation. In the continuous case, if we define the Bayes factor for ψ as a limit as in (Equation 1), then this limit equals $RB_{\Psi }\left(\psi \;|\;x\right).$ Further discussion on the choice of a measure of evidence can be found in [2] as there are other candidates beyond these two. One significant advantage for the relative belief ratio is that all inferences derived based on it are invariant under smooth reparameterizations. Furthermore, the relative belief ratio only serves to order the values of ψ∈Ψ with respect to evidence, and the value $RB_{\Psi }\left(\psi \;|\;x\right)$ is not to be considered as measuring evidence on a universal scale. It is important to note that the discussion of bias here depends only on the principle of evidence and is the same no matter what valid measure of evidence is used.

Since the model and prior are subjectively chosen, the characterization and measurement of statistical evidence has a subjective component. This creates the possibility that these choices are biased, namely, they were chosen with some goal in mind other than letting the data determine the conclusions. Model checking and checking for prior-data conflict exposes these choices to criticism via the data, but these checks will not reveal inappropriate conduct like tailoring a model or prior based on the observed data. Perhaps a more important check on such behavior is to measure and control bias. As will now be shown, controlling the bias through the a priori determination of the amount of data collected can leave us with greater confidence that the data are the primary driver of whatever inferences are drawn, and this is surely the goal in scientific applications. Thus, while informed subjective choices are a good thing, there are also tools that can be used to mitigate concerns about subjectivity, as these allow an analysis to at least approach the scientific goal of an objective analysis. The lack of a precise definition of objectivity, and a clear methodology for attaining it, is not a failure since the issue can be addressed. This is a somewhat nuanced view of the objective/subjective concern and is perhaps more in line with the views on this topic as expressed in [25,26].

### 3.1. Bias in Hypothesis Assessment Problems

Suppose the problem of interest is to assess whether or not there is evidence in favor of or against $H_{0}:\Psi \left(\theta \right)=\psi_{*},$ as is determined here by $RB_{\Psi }\left(\psi_{*}\;|\;x\right)$ being greater than or less than 1. It is to be noted that no restrictions, beyond propriety, are placed on priors here so Π could very well be a mixture of a prior on $H_{0}\subset \Theta$ and a prior on $H_{0}^{c}$ with $H_{0}$ assigned some positive mass as is commonly done in Bayesian testing problems. Certainly, such a prior is necessary when $\Psi =I_{H_{0}}$ and $\psi_{*}=1$ so the relevant relative belief ratio is $RB(H_{0}\;|\;x).$ While this formulation is accommodated, there is no reason to insist that every hypothesis assessment be expressed this way. When Ψ(θ) is a quantity like a mean, variance, quantile, etc., it seems natural to compare the value $RB_{\Psi }\left(\psi_{*}\;|\;x\right)$ with each of the other possible values $RB_{\Psi }\left(\psi \;|\;x\right)$ for ψ∈Ψ to calibrate, as is done subsequently via (Equation 2), how strong the evidence is concerning $\psi_{*}.$

The following example is carried along as it illustrates a number of things.

**Example** **3.**
*Location normal.*


Suppose $x=(x_{1},\ldots ,x_{n})$ is i.i.d. $N(\mu ,\sigma_{0}^{2})$ with π a $N(\mu_{0},\tau_{0}^{2})$ prior. Then, $\mu \;|\;x∼N({\left(n/\sigma_{0}^{2}+1/\tau_{0}^{2}\right)}^{-1}\left(n\bar{x}/\sigma_{0}^{2}+\mu_{0}/\tau_{0}^{2}\right),{\left(n/\sigma_{0}^{2}+1/\tau_{0}^{2}\right)}^{-1})$ and so RB(μ|x) equals
$${\left(1+\frac{n\tau_{0}^{2}}{\sigma_{0}^{2}}\right)}^{1/2}\exp\left\{-\frac{1}{2}{\left(1+\frac{\sigma_{0}^{2}}{n\tau_{0}^{2}}\right)}^{-1}{\left(\frac{\sqrt{n}\left(\bar{x}-\mu \right)}{\sigma_{0}}+\frac{\sigma_{0}\left(\mu_{0}-\mu \right)}{\sqrt{n}\tau_{0}^{2}}\right)}^{2}+\frac{{\left(\mu -\mu_{0}\right)}^{2}}{2\tau_{0}^{2}}\right\}.$$

Observe that, as $\tau_{0}^{2}\to \infty$, then RB(μ|x)→∞ for every μ and in particular for a hypothesized value $H_{0}=\left\{\mu_{*}\right\}.$ Thus, it would appear that overwhelming evidence is obtained for the hypothesis when the prior is very diffuse, and this holds irrespective of what the data says. In addition, when the standardized value $\sqrt{n}\left|\bar{x}-\mu_{*}\right|/\sigma_{0}$ is fixed, then $RB(\mu_{*}\;|\;x)\to \infty$ as n→∞. These phenomena also occur if a Bayes factor (which equals $RB(\mu_{*}\;|\;x)$ in this case) or a posterior probability based upon a discrete prior mass at $\mu_{*}$, are used to assess $H_{0}.$ Accordingly, all these measures lead to a sharp disagreement with the frequentist *p*-value $2(1-\Phi (\sqrt{n}|\bar{x}-\mu_{*}|/\sigma_{0}))$ when it is small. This is the Jeffreys–Lindley paradox, and it arises quite generally.

The Jeffreys–Lindley paradox shows that the strength of evidence cannot be measured strictly by the size of the measure of evidence. A logical way to assess strength is to compare the evidence for $\psi_{*}$ with the evidence for the other values for ψ. The *strength* can then be measured by
(2)$$\Pi_{\Psi }\left(RB_{\Psi }\left(\psi \;|\;x\right)\leq RB_{\Psi }\left(\psi_{*}\;|\;x\right)\;|\;x\right),$$
the posterior probability that the true value has evidence no greater than the evidence for $\psi_{*}.$ Thus, if $RB_{\Psi }\left(\psi_{*}\;|\;x\right)<1$ and (Equation 2) is small, then there is strong evidence against $\psi_{*}$, while, if $RB_{\Psi }\left(\psi_{*}\;|\;x\right)>1$ and (Equation 2) is large, then there is strong evidence in favor of $\psi_{*}.$ The inequalities $\Pi_{\Psi }\left(\left\{\psi_{*}\right\}\;|\;x\right)\leq \Pi_{\Psi }\left(RB_{\Psi }\left(\psi \;|\;x\right)\leq RB_{\Psi }\left(\psi_{*}\;|\;x\right)\;|\;x\right)\leq RB_{\Psi }\left(\psi_{*}\;|\;x\right)$ hold and thus, when $RB_{\Psi }\left(\psi_{*}\;|\;x\right)$ is small, there is strong evidence against $\psi_{*}$ and, when $RB_{\Psi }\left(\psi_{*}\;|\;x\right)>1$ and $\Pi_{\Psi }\left(\left\{\psi_{*}\right\}\;|\;x\right)$ is big, then there is strong evidence in favor of $\psi_{*}.$ Note, however, that $\Pi_{\Psi }\left(\left\{\psi_{*}\right\}\;|\;x\right)\approx 1$ does not guarantee $RB_{\Psi }\left(\psi_{*}\;|\;x\right)>1$ and, if $RB_{\Psi }\left(\psi_{*}\;|\;x\right)<1$, this means that there is weak evidence against $\psi_{*}.$ In addition, there is no reason why multiple measures of the strength of the evidence can’t be used (see the discussion in Section 3.2). In fact, when Ψ is binary-valued, it is better to use $\Pi_{\Psi }\left(\left\{\psi_{*}\right\}\;|\;x\right)$ to measure the strength, as we did in Examples 1 and 2, and there are also some issues with (Equation 2) in the continuous case that can require a modification. These issues are ignored here, as the strength does not play a role when considering bias, and the reader can see [2] for further discussion. The important point is that it is necessary to calibrate the measure of evidence using probability to measure how strong belief in the evidence is and (Equation 2) is a reasonable way to do this in many contexts.

***1.*** Example 3 *Location normal (continued).*

A simple calculation shows that, with $\sqrt{n}\left|\bar{x}-\mu_{*}\right|$ fixed, (Equation 2) then converges to $2(1-\Phi (\sqrt{n}|\bar{x}-\mu_{*}|/\sigma_{0}))$ as $n\tau_{0}^{2}\to \infty .$ Thus, if the *p*-value is small, this indicates that a large value of $RB_{\Psi }\left(\mu_{*}\;|\;x\right)$ is only weak evidence in favor of $\mu_{*}.$ It is to be noted that the *p*-value $2(1-\Phi (\sqrt{n}|\bar{x}-\mu_{*}|/\sigma_{0}))$ is not a valid measure of evidence as described here because there is no cut-off that corresponds to evidence for and evidence against. Thus, its appearance as a measure of the strength of the evidence is not circular.

Simple algebra shows (see the Appendix A), however, that
$$\begin{array}{cc} \; & 2(1-\Phi (\sqrt{n}|\bar{x}-\mu_{*}|/\sigma_{0}))-\\ \; & 2(1-\Phi \left({\left[\log\left(1+n\tau_{0}^{2}/\sigma_{0}^{2}\right)+{\left(1+\sigma_{0}^{2}/n\tau_{0}^{2}\right)}^{-1}{\left(\bar{x}-\mu_{0}\right)}^{2}/\tau_{0}^{2}\right]}^{1/2}\right), \end{array}$$
a difference of two *p*-values, is a valid measure of evidence via the cut-off 0. From this, it is seen that the values of the first *p*-value $2(1-\Phi (\sqrt{n}|\bar{x}-\mu_{*}\left|/\sigma_{0}\right)$ that lead to evidence against, generally become smaller as $n\tau_{0}^{2}\to \infty .$ For example, with $n=10,\sigma_{0}^{2}=1,\mu_{*}=0$ and $\sqrt{n}\left|\bar{x}-\mu_{*}\right|/\sigma_{0}=1.96,$ the standard *p*-value equals 0.05. Setting $\mu_{0}=0$ and $\tau_{0}^{2}=1$, the second *p*-value equals 0.097 and thus there is evidence against $\mu_{*}=0$, with $\tau_{0}^{2}=10$ being the second term equal to 0.031 and, with $\tau_{0}^{2}=100$, it equals 0.009, so there is evidence in favor of $\mu_{*}=0$ in both cases. When *n* increases, these values become smaller, as, with n=50, the first *p*-value equal to 0.05 is always evidence in favor. Similar results are obtained with a uniform prior on (−m,m), reflecting perhaps a desire to treat many values equivalently, as m→∞ or n→∞. For example, with m=10 and $n=10,\sigma_{0}^{2}=1$, $\mu_{*}=0,\sqrt{n}\left|\bar{x}-\mu_{*}\right|/\sigma_{0}=1.96,$ then the second *p*-value equals 0.002, and there is evidence in favor of $\mu_{*}=0.$ These findings are similar to those in [27,28].

It is very simple to elicit $\left(\mu_{0},\tau_{0}^{2}\right)$ based on prescribing an interval that contains the true μ with some high probability such as 99.9%, taking $\mu_{0}$ to be the mid-point and so $\tau_{0}^{2}$ is determined. There is no reason to take $\tau_{0}^{2}$ to be arbitrarily large. However, one still wonders if the choice made is inducing some kind of bias into the problem as taking $\tau_{0}^{2}$ too large clearly does.

Certainly, default choices of priors should be avoided when possible, but even when eliciting, how can we know if the chosen prior is inducing bias? To assess this, a numerical measure is required. The principle of evidence suggests that *bias against*
$H_{0}$ is measured by
(3)$$M(RB_{\Psi }\left(\psi_{*}\;|\;X\right)\leq 1\;|\;\psi_{*})$$
where $M(\cdot \;|\;\psi_{*})$ is the prior predictive distribution of the data given that the hypothesis is true. Thus, (Equation 3) is the prior probability that evidence in favor of $\psi_{*}$ will not be obtained when $\psi_{*}$ is the true value. If (Equation 3) is large, then there is an a priori bias against $H_{0}.$

For the bias in favor of $H_{0}$, it is necessary to assess if evidence against $H_{0}$ will not be obtained with high prior probability even when $H_{0}$ is false. One possibility is to measure *bias in favor* by
(4)$$\begin{array}{cc} \; & \int_{\Psi \backslash \{\psi_{*}\}}M\left(RB_{\Psi }\left(\psi_{*}\;|\;X\right)\geq 1\;|\;\psi \right)\;\Pi_{\Psi }\left(d\psi \right)\\ \; & =M\left(RB_{\Psi }\left(\psi_{*}\;|\;X\right)\geq 1\right)-M\left(RB_{\Psi }\left(\psi_{*}\;|\;X\right)\geq 1\;|\;\psi_{*}\right)\Pi_{\Psi }\left(\left\{\psi_{*}\right\}\right), \end{array}$$
the prior probability of not obtaining evidence against $\psi_{*}$ when it is false. When $\Pi_{\Psi }\left(\left\{\psi_{*}\right\}\right)=0,$ (Equation 4) equals $M(RB_{\Psi }\left(\psi_{*}\;|\;X\right)\geq 1)$, where *M* is the prior predictive for the data. For continuous parameters, it can be argued that it doesn’t make sense to consider values of ψ so close to $\psi_{*}\;$that they are practically indistinguishable. Suppose that there is a measure of distance $d_{\Psi }$ on Ψ and a value δ>0 such that, if $d_{\Psi }\left(\psi_{*},\psi \right)<\delta ,$ then $\psi_{*}$ and ψ are indistinguishable in the application. The *bias in favor* of $H_{0}$ is then measured by replacing $\Psi \backslash \{\psi_{*}\}$ in (Equation 4) by $\left\{\psi :d_{\Psi }\left(\psi_{*},\psi \right)\geq \delta \right\}$ leading to the upper bound
(5)$$\underset{\left\{\psi :d_{\Psi }\left(\psi_{*},\psi \right)\geq \delta \right\}}{\sup}M\left(RB_{\Psi }\left(\psi_{*}\;|\;X\right)\geq 1\;|\;\psi \right).$$

Typically, $M(RB_{\Psi }\left(\psi_{*}\;|\;X\right)\geq 1\;|\;\psi )$ decreases as ψ moves away from $\psi_{*}$ so (Equation 5) can be computed by finding the supremum over the set $\left\{\psi :d_{\Psi }\left(\psi_{*},\psi \right)=\delta \right\}$ and, when ψ is real-valued and $d_{\Psi }$ is Euclidian distance, this set equals $\{\psi_{*}-\delta ,\psi_{*}+\delta \}.$

It is to be noted that the measures of bias given by (Equation 3)–(Equation 5) do not depend on using the relative belief ratio to measure evidence. Any valid measure of evidence will determine the same values when the relevant cut-off is substituted for 1. It is only (Equation 2) that depends on the specific choice of the relative belief ratio as the measure of evidence.

Under general circumstances, see [2], both biases will converge to 0 as the amount of data increases and thus they can be controlled by the amount of data collected. There is no point in reporting the results of an analysis when there is a lot of bias unless the evidence contradicts the bias.

***2.*** Example 3 *Location normal (continued).*

Under M(·|μ), then $\bar{x}∼N\left(\mu ,\tau_{0}^{2}+\sigma_{0}^{2}/n\right).\;$Thus, putting
$$\begin{array}{cc} \; & a\left(\mu ,\mu_{0},\tau_{0}^{2},\sigma_{0}^{2},n\right)=\sigma_{0}\left(\mu -\mu_{0}\right)/\sqrt{n}\tau_{0}^{2},\\ \; & b\left(\mu ,\mu_{0},\tau_{0}^{2},\sigma_{0}^{2},n\right)={\left\{\left(1+\sigma_{0}^{2}/n\tau_{0}^{2}\right)\left[\log\left(1+n\tau_{0}^{2}/\sigma_{0}^{2}\right)+{\left(\mu -\mu_{0}\right)}^{2}/\tau_{0}^{2}\right]\right\}}^{1/2}, \end{array}$$
then (Equation 3) is given by
(6)$$\begin{array}{cc} M(RB(\mu \;|\;X)\leq 1\;|\;\mu )=1- & \Phi \left(a\left(\mu ,\mu_{0},\tau_{0}^{2},\sigma_{0}^{2},n\right)+b\left(\mu ,\mu_{0},\tau_{0}^{2},\sigma_{0}^{2},n\right)\right)+\\ \; & \Phi \left(a\left(\mu ,\mu_{0},\tau_{0}^{2},\sigma_{0}^{2},n\right)-b\left(\mu ,\mu_{0},\tau_{0}^{2},\sigma_{0}^{2},n\right)\right). \end{array}$$

This goes to 0 as n→∞ or as $\tau_{0}^{2}\to \infty .$ Thus, bias against can be controlled by sample size *n* or by the diffuseness of the prior although, as subsequently shown, a diffuse prior induces bias in favor. It is also the case that (Equation 6) converges to 0 when $\mu_{0}\to \pm \infty \;$or when $\sigma_{0}/\sqrt{n}\tau_{0}$ is fixed and $\tau_{0}\to 0.$ Thus, it would appear that using a prior with a location quite different than the hypothesized value or a prior that was much more concentrated than the sampling distribution can be used to lower bias against. These are situations, however, where one can expect to have prior-data conflict after observing the data.

The entries in Table 2 record the bias against for a specific case and illustrate that increasing *n* does indeed reduce bias. The entries also show that bias against can be greater when the prior is centered on the hypothesis. Figure 1 contains a plot of the bias against $H_{0}=\left\{\mu \right\},$ as a function of μ, when using a N(0,1) prior. Note that the maximum bias against occurs at the mean of the prior (and equals 0.143), and this typically occurs when $\sigma_{0}^{2}/n\tau_{0}^{2}<1,$ namely, when the data are more concentrated than the prior. Figure 1 also contains a plot of the bias against when using a prior more concentrated than the data distribution. That the bias against is maximized, as a function of the hypothesized mean μ, when μ equals the value associated with the strongest belief under the prior, seems odd. This phenomenon arises quite often, and the mathematical explanation for this is that the greater the amount of prior probability assigned to a value, the harder it is for the posterior probability to increase and so it is quite logical when considering evidence. It will be seen that this phenomenon is very convenient for the control of bias in estimation problems and could be used as an argument for using a prior centered on the hypothesis, although this is not necessary as beliefs may be different.

Now, consider (Equation 5), namely, bias in favor of $H_{0}=\left\{\mu_{*}\right\}.$ Putting
$$c\left(\mu_{*},\mu ,\mu_{0},\tau_{0}^{2},\sigma_{0}^{2},n\right)=\sqrt{n}\left(\mu_{*}-\mu \right)/\sigma_{0}+a\left(\mu_{*},\mu_{0},\tau_{0}^{2},\sigma_{0}^{2},n\right),$$
then (Equation 5) equals $\max M(RB\left(\mu_{*}\;|\;X\right)\geq 1\;|\;\mu_{*}\pm \delta )$ where
(7)$$\begin{array}{cc} M(RB\left(\mu_{*}\;|\;X\right)\geq 1\;|\;\mu )=\; & \Phi \left(c\left(\mu_{*},\mu ,\mu_{0},\tau_{0}^{2},\sigma_{0}^{2},n\right)+b\left(\mu_{*},\mu_{0},\tau_{0}^{2},\sigma_{0}^{2},n\right)\right)-\\ \; & \Phi \left(c\left(\mu_{*},\mu ,\mu_{0},\tau_{0}^{2},\sigma_{0}^{2},n\right)-b\left(\mu_{*},\mu_{0},\tau_{0}^{2},\sigma_{0}^{2},n\right)\right) \end{array}$$
which converges to 0 as n→∞ and also as μ→±∞. However, (Equation 7) converges to 1 as $\tau_{0}^{2}\to \infty ,$ so, if the prior is too diffuse, there will be bias in favor of $\mu_{*}.$ Thus, resolving the Jeffreys–Lindley paradox requires choosing the sample size *n*, after choosing the prior, so that (Equation 7) is suitably small. Note that choosing $\tau_{0}^{2}$ to be larger reduces bias against but increases bias in favor and so generally bias cannot be avoided by choice of prior. Figure 2 is a plot of $M(RB\left(\mu_{*}\;|\;X\right)\geq 1\;|\;\mu )$ for a particular case and this strictly decreases as μ moves away from $\mu_{*}$.

In Table 3, we have recorded some specific values of the bias in favor using (Equation 4) and using (Equation 5) where $d_{\Psi }$ is Euclidean distance. It is seen that bias in favor can be quite serious for small samples. When using (Equation 5), this can be mitigated by making δ larger. For example, with $\left(\mu_{0},\tau_{0}\right)=\left(0,1\right),\delta =1.0,n=20$, the bias in favor equals 0.004. Note, however, that δ is not chosen to make the bias in favor small; rather, it is determined in an application as the difference from the null that is just practically important. The virtues of a suitable value of δ are readily apparent as (Equation 5) is much smaller than (Equation 4) for larger n.

A comparison of Table 2 and Table 3 shows that a study whose purpose is to demonstrate evidence in favor of $H_{0}$ is much more demanding than one whose purpose is to determine whether or not there is evidence against $H_{0}.$ As a cautionary note too, it is worth reminding the reader that bias is not to be used in the selection of a prior. The prior is to be selected by elicitation and the biases measured for that prior. If one or both biases are too large, then that is telling us that more data are needed to ensure that the conclusions drawn are primarily driven by the data and not the prior. It is tempting to look at Table 2 and Table 3 and compare the priors, but this is not the way to proceed and it can be seen that choosing a prior to minimize one bias simply increases the other. It is also the case that bias can be measured when a default proper prior is chosen, see Example 3, as is often done when considering sparsity inducing priors, but the discussion here will focus on the ideal where elicitation can be carried out. One can argue that bias is also model dependent and that is certainly true so, while our focus is on the prior, in reality, the biases are a measure of the model-prior combination. The same comment applies to the model, however, that bias measurements are not to be used to select a model.

### 3.2. The Role of the Difference that Matters δ

The role and value of δ require some further discussion as some may find the need to specify this quantity controversial. The value of δ depends on the application as well as the characteristic of interest ψ=Ψ(θ). For the developments here, specifying δ is a necessary part of the investigation. There may well be contexts where the precise value of δ is unclear. That seems to suggest, however, that the investigator does not fully understand what ψ is as a real-world object and formal inference in such a context seems questionable, although perhaps some kind of exploratory analysis is reasonable. In a well-designed study, a measurement process is selected which, together with sampling from the population, determines the data. In deciding on the measurement process, and sample size, an investigator has to decide on the accuracy required and that is where δ enters the picture.

Consider a problem where an investigator is measuring the length of some quantity associated with each member of a population and wants to make inferences about the mean length ψ. If the investigator chooses to measure each length to the nearest cm, then there is no way that the true value of the mean can be known to an accuracy beyond ±0.5 cm, even if the entire population is measured. As another example, suppose that ψ represents the proportion of individuals in a population infected with a virus. Surely, it is imperative to settle on how accurately we wish to know ψ and that will play a key role in a number of statistical activities like determining sample size for the consideration of a hypothesis concerning the true value of ψ. For example, does the application require that ψ be known within an absolute error of δ or within a relative error of δ? See [29] for discussion on this point in the context of logistic regression. To simply proceed to collect data and do a statistical analysis without taking such considerations into account does not seem like good practice.

While discussion of δ may be limited, it has certainly not disappeared from the statistical literature. For example, consider power studies where a δ is required. In addition, one of the many criticisms of the *p*-value arises because, for a large enough sample size, a difference may be detected that is of no importance. The general recommendation is to then quote a confidence interval to see if that is the case, but it is difficult to see how that is helpful unless one knows what difference δ matters. This has long been an issue when discussing testing problems, see [30], and yet it still seems unresolved as it is not always clear how to obtain an appropriate *p*-value that incorporates δ. One of the benefits of the approach here is that it is straightforward to incorporate δ into the analysis and, in fact, it often makes an analysis easier. Thus, specifying δ is a part of every well-designed statistical investigation.

### 3.3. Bias in Estimation Problems

The relative belief estimate of ψ=Ψ(θ) is the value that maximizes the measure of evidence, namely, $\psi \left(x\right)=\arg\sup RB_{\Psi }\left(\psi \;|\;x\right).$ It is easy to show that $RB_{\Psi }\left(\psi \left(x\right)\;|\;x\right)\geq 1$ with the inequality strict except in trivial contexts. The accuracy of this estimate can be measured by the “size” of the *plausible region*$Pl_{\Psi }\left(x\right)=\left\{\psi :RB_{\Psi }\left(\psi \;|\;x\right)>1\right\},$ the set of values of ψ that have evidence in their favor and note $\psi \left(x\right)\in Pl_{\Psi }\left(x\right).$ To say that ψ(x) is an accurate estimate requires that $Pl_{\Psi }\left(x\right)$ be “small”, perhaps as measured by $Vol(Pl_{\Psi }\left(x\right))$, where Vol is some measure of volume, and also has high posterior content $\Pi_{\Psi }\left(Pl_{\Psi }\left(x\right)\;|\;x\right),$ which measures the belief that the true value is in $Pl_{\Psi }\left(x\right).$ Note that $Pl_{\Psi }\left(x\right)$ does not depend on the specific measure of evidence chosen, in this case the relative belief ratio. Any valid estimator must satisfy the principle of evidence and thus be in $Pl_{\Psi }\left(x\right).$ It is now argued that, in an estimation problem, bias is measured by various coverage probabilities for the plausible region.

Note too that, if there is evidence in favor of $H_{0}:\Psi \left(\theta \right)=\psi_{*},$ then $\psi_{*}\in Pl_{\Psi }\left(x\right)$ and so represents the natural estimate of ψ provided there was a clear reason, like the assessment of a scientific theory, for assessing the evidence for this value. This assumes too that there isn’t substantial bias in favor of $\psi_{*}$. The strength of the evidence in favor of $\psi_{*}$ could then also be measured by the size of $Pl_{\Psi }\left(x\right).$ Similarly, if evidence against $H_{0}$ is obtained, then $\psi_{*}\in Im_{\Psi }\left(x\right)=\left\{\psi :RB_{\Psi }\left(\psi \;|\;x\right)<1\right\}$ the *implausible region*, and there is strong evidence against $H_{0}$ provided $Im_{\Psi }\left(x\right)$ has small volume and large posterior probability. A virtue of this approach to measuring the strength of the evidence is that it does not depend upon using the relative belief ratio in hypothesis assessment problems.

The prior probability that the plausible region does not cover the true value measures bias against when estimating ψ. If this probability is large, then the estimate and the plausible region are a priori likely to be misleading as to the true value. The prior probability that $Pl_{\Psi }\left(x\right)$ doesn’t contain ψ=Ψ(θ) when $\theta ∼\Pi ,X∼P_{\theta }$ is
(8)$$E_{\Pi_{\Psi }}\left(M(\psi \notin Pl_{\Psi }\left(X\right)\;|\;\psi )\right)=E_{\Pi_{\Psi }}\left(M\left(RB_{\Psi }\left(\psi \;|\;X\right)\leq 1\;|\;\psi \right)\right)$$
which is also the average bias against over all hypothesis testing problems $H_{0}:\Psi \left(\theta \right)=\psi .$ Note $1-E_{\Pi_{\Psi }}\left(M(\psi \notin Pl_{\Psi }\left(X\right)\;|\;\psi )\right)=E_{\Pi_{\Psi }}\left(M(\psi \in Pl_{\Psi }\left(X\right)\;|\;\psi )\right)=E_{M}\left(\Pi_{\Psi }\left(Pl_{\Psi }\left(X\right)\;|\;X\right)\right)$ which is the prior coverage probability of $Pl_{\Psi }$. In addition,
(9)$$\underset{\psi }{\sup}M\left(\psi \notin Pl_{\Psi }\left(X\right)\;|\;\psi \right)=\underset{\psi }{\sup}M\left(RB_{\Psi }\left(\psi \;|\;X\right)\leq 1\;|\;\psi \right),$$
is an upper bound on (Equation 8). Therefore, controlling (Equation 9) controls the bias against in estimation and all hypothesis assessment problems involving ψ. In addition,
$$1-\underset{\psi }{\sup}M\left(\psi \notin Pl_{\Psi }\left(X\right)\;|\;\psi \right)=\underset{\psi }{\inf}M\left(\psi \in Pl_{\Psi }\left(X\right)\;|\;\psi \right)\leq E_{M}\left(\Pi_{\Psi }\left(Pl_{\Psi }\left(X\right)\;|\;X\right)\right).$$

Thus, using (Equation 9) implies lower bounds for the coverage probability and for the expected posterior content of the plausible region. In general, both (Equation 8) and (Equation 9) converge to 0 with increasing amounts of data. Thus, it is possible to control for bias against in estimation problems by the amount of data collected.

***3.*** Example 3 *Location normal (continued).*

The value of M(RB(μ|X)≤1|μ) is given in (Equation 6) and examples are plotted in Figure 1. When $\mu ∼N(\mu_{0},\tau_{0}^{2})$, then $z=\left(\mu -\mu_{0}\right)/\tau_{0}∼N\left(0,1\right)$, so
$$\begin{array}{cc} \; & E_{\Pi }\left(M(RB(\mu \;|\;X)\leq 1\;|\;\mu )\right)=\\ \; & 1-E\left[\begin{array}{c} \Phi \left(\frac{\sigma_{0}}{\sqrt{n}\tau_{0}}Z+{\left\{\left(1+\frac{\sigma_{0}^{2}}{n\tau_{0}^{2}}\right)\left[\log\left(1+\frac{n\tau_{0}^{2}}{\sigma_{0}^{2}}\right)+Z^{2}\right]\right\}}^{1/2}\right)+\\ \Phi \left(\frac{\sigma_{0}}{\sqrt{n}\tau_{0}}Z-{\left\{\left(1+\frac{\sigma_{0}^{2}}{n\tau_{0}^{2}}\right)\left[\log\left(1+\frac{n\tau_{0}^{2}}{\sigma_{0}^{2}}\right)+Z^{2}\right]\right\}}^{1/2}\right) \end{array}\right] \end{array}$$
which is notably independent of the prior mean $\mu_{0}$. The dominated convergence theorem implies $E_{\Pi }\left(M(RB(\mu \;|\;X)\leq 1\;|\;\mu )\right)\to 0$ as n→∞ or as $\tau_{0}^{2}\to \infty .$ Thus, provided $n\tau_{0}^{2}/\sigma_{0}^{2}$ is large enough, there is hardly any estimation bias against. Table 4 illustrates some values of this bias measure. Subtracting the probabilities in Table 4 from 1 gives the prior probability that the plausible region covers the true value and the expected posterior content of the plausible region. Thus, when $n=20,\tau_{0}=1,$ the prior probability of Pl(x) containing the true value is 1−0.051=0.949 so Pl(x) is a 0.949 Bayesian confidence interval for μ.

To use (Equation 9), it is necessary to maximize M(RB(μ|X)≤1|μ) as a function of μ and it is seen that, at least when the prior is not overly concentrated, this maximum occurs at $\mu_{0}.$
Figure 1 shows that, when using the N(0,1) prior, the maximum occurs at μ=0 when n=5 and, from the second column of Table 2, the maximum equals 0.143. The average bias against is given by 0.107, as recorded in Table 4. Note that the maximum also occurs at μ=0 for the other values of *n* recorded in Table 2.

Bias in favor when estimating ψ occurs when the prior probability that $Im_{\Psi }$ does not cover a false value is large, namely, when
(10)$$\begin{array}{cc} \; & \int_{\Psi }\int_{\Psi \backslash \{\psi_{*}\}}M\left(\psi_{*}\notin Im_{\Psi }\left(X\right)\;|\;\psi \right)\;\Pi_{\Psi }\left(d\psi \right)\;\Pi_{\Psi }\left(d\psi_{*}\right)\\ \; & =\int_{\Psi }\int_{\Psi \backslash \{\psi_{*}\}}M\left(RB_{\Psi }\left(\psi_{*}\;|\;X\right)\geq 1\;|\;\psi \right)\;\Pi_{\Psi }\left(d\psi \right)\;\Pi_{\Psi }\left(d\psi_{*}\right) \end{array}$$
is large as this would seem to imply that the plausible region will cover a randomly selected false value from the prior with high prior probability. Note that (Equation 10) is the prior mean of (Equation 4) and, in the continuous case, equals $\int_{\Psi }M\left(\psi_{*}\notin Im_{\Psi }\left(X\right)\right)\;\Pi_{\Psi }\left(d\psi_{*}\right)$. As previously discussed, however, it often doesn’t make sense to distinguish values of ψ that are close to $\psi_{*}.$ The bias in favor for estimation can then be measured by
(11)$$\begin{array}{cc} & E_{\Pi_{\Psi }}\left(\underset{\left\{\psi :d_{\Psi }\left(\psi ,\psi_{*}\right)\geq \delta \right\}}{\sup}M\left(\psi_{*}\notin Im_{\Psi }\left(X\right)\;|\;\psi \right)\right)\\ = & E_{\Pi_{\Psi }}\left(\underset{\left\{\psi :d_{\Psi }\left(\psi ,\psi_{*}\right)\geq \delta \right\}}{\sup}M\left(RB_{\Psi }\left(\psi_{*}\;|\;X\right)\geq 1\;|\;\psi \right)\right). \end{array}$$

An upper bound on (Equation 11) is commonly equal to 1, as illustrated in Figure 3, and thus is not useful.

It is the size and posterior content of $Pl_{\Psi }\left(x\right)$ that provides a measure of the accuracy of the estimate ψ(x). As previously discussed, the a priori expected posterior content of $Pl_{\Psi }\left(x\right)$ can be controlled by bias against. The a priori expected volume of $Pl_{\Psi }\left(x\right)$ satisfies
(12)$$E_{M}\left(Vol(Pl_{\Psi }\left(X\right))\right)=\int_{\Psi }\int_{\Psi }M\left(\psi_{*}\in Pl_{\Psi }\left(X\right)\;|\;\psi \right)\;\Pi_{\Psi }\left(d\psi \right)\;Vol\left(d\psi_{*}\right).$$

Notice that, when $\Pi_{\Psi }\left(\left\{\psi \right\}\right)=0$ for every ψ, this can be interpreted as a kind of average of the prior probabilities of the plausible region covering a false value.

***4.*** Example 3 *Location normal (continued).*

It follows from (Equation 7) that
$$\begin{array}{cc} \; & \sup M(RB\left(\mu_{*}\;|\;X\right)\geq 1\;|\;\mu_{*}\pm \delta )\;=\;\\ \; & \sup\left\{\begin{array}{c} \Phi \left(c\left(\mu_{*},\mu_{*}\pm \delta ,\mu_{0},\tau_{0}^{2},\sigma_{0}^{2},n\right)+b\left(\mu_{*},\mu_{0},\tau_{0}^{2},\sigma_{0}^{2},n\right)\right)-\\ \Phi \left(c\left(\mu_{*},\mu_{*}\pm \delta ,\mu_{0},\tau_{0}^{2},\sigma_{0}^{2},n\right)-b\left(\mu_{*},\mu_{0},\tau_{0}^{2},\sigma_{0}^{2},n\right)\right) \end{array}\right\} \end{array}$$

Note that, as $\mu_{*}\to \pm \infty$, then $M(RB\left(\mu_{*}\;|\;X\right)\geq 1\;|\;\mu_{*}\pm \delta )\to 1$ when $n\tau_{0}^{2}/\sigma_{0}^{2}>1,$ see Figure 3, and converges to 0 if $n\tau_{0}^{2}/\sigma_{0}^{2}<1,$ so it would appear that the better circumstance for guarding against bias in favor is when the prior is putting in more information than the data. As previously noted, however, this is a situation where we might expect prior data-conflict to arise and, except in exceptional circumstances, should be avoided. Table 5 contains values of (Equation 11) for this situation with different values of δ. Again, these values are just for illustrative purposes and are not to be used to compare or choose priors.

Some elementary calculations give $Pl\left(x\right)=\bar{x}\pm w\left(\bar{x},n,\sigma_{0}^{2},\mu_{0},\tau_{0}^{2}\right)$ with
$$w\left(\bar{x},n,\sigma_{0}^{2},\mu_{0},\tau_{0}^{2}\right)=\frac{\sigma_{0}}{\sqrt{n}}{\left(1+\frac{n\tau_{0}^{2}}{\sigma_{0}^{2}}\right)}^{-\frac{1}{2}}{\left\{\left(1+\frac{n\tau_{0}^{2}}{\sigma_{0}^{2}}\right)\log\left(1+\frac{n\tau_{0}^{2}}{\sigma_{0}^{2}}\right)+{\left(\frac{\bar{x}-\mu_{0}}{\sigma_{0}/\sqrt{n}}\right)}^{2}\right\}}^{\frac{1}{2}}$$
where $z=\sqrt{n}\left(\bar{x}-\mu_{0}\right)/\sigma_{0}∼N\left(0,1\right)$ under M. It is notable that the prior distribution of the width is independent of the prior mean. Table 6 contains some expected half-widths together with the coverage probabilities of Pl(x).

While the plausible region $Pl_{\Psi }\left(x\right)$ is advocated for assessing the accuracy of estimates, it is also possible to use a γ−relative belief credible region $C_{\gamma }\left(x\right)=\left\{\psi :RB_{\Psi }\left(\psi \;|\;x\right)\geq c_{\gamma }\left(x\right)\right\}\;$ where $c_{\gamma }\left(x\right)=\inf\left\{c:\Pi_{\Psi }\left(RB_{\Psi }\left(\psi \;|\;x\right)\geq c\;|\;x\right)\leq \gamma \right\}.$ There is one proviso with this, however, as the principle of evidence requires that $\gamma \leq \Pi_{\Psi }\left(Pl_{\Psi }\left(x\right)\;|\;x\right);$ otherwise, $C_{\gamma }\left(x\right)$ will contain values of ψ for which there is evidence against. Notice that, while controlling the bias against allows control of the coverage probability of $Pl_{\Psi }\left(x\right)$, this does not control the coverage probability of a credible region since $\Pi_{\Psi }\left(Pl_{\Psi }\left(x\right)\;|\;x\right)$ is not known until the data are observed. For this reason, reporting the plausible region always seems necessary. All these regions are invariant under smooth reparameterizations and in [31] various optimality results are established for these credible regions.

## 4. Frequentist and Optimal Properties

Consider now the bias against $H_{0}=\left\{\psi_{*}\right\},$ namely, $M(RB_{\Psi }\left(\psi_{*}\;|\;X\right)\leq 1\;|\;\psi_{*}).$ If we repeatedly generate $\theta ∼\pi \left(\cdot \;|\;\psi_{*}\right),X∼f_{\theta },$ then this probability is the long-run proportion of times that $RB_{\Psi }\left(\psi_{*}\;|\;X\right)\leq 1.$ This frequentist interpretation depends on the conditional prior $\pi (\cdot \;|\;\psi_{*})\;$and, when Ψ(θ)=θ, there are no nuisance parameters, this is a “pure” frequentist probability. Even in the latter case, there is some dependence on the prior, however, as $RB\left(\theta_{*}\;|\;x\right)=f_{\theta_{*}}\left(x\right)/m\left(x\right)$ so *x* satisfies $RB_{\Psi }\left(\theta_{*}\;|\;x\right)\leq 1$ iff $f_{\theta_{*}}\left(x\right)\leq m\left(x\right)$, where $m\left(x\right)=\int_{\Theta }f_{\theta }\left(x\right)\;\Pi \left(d\theta \right).$ Thus, in general, the region $\left\{x:RB_{\Psi }\left(\psi_{*}\;|\;x\right)\leq 1\right\}$ depends on π, but the probability $M(RB_{\Psi }\left(\psi_{*}\;|\;X\right)\leq 1\;|\;\psi_{*})$ depends only on the conditional prior predictive given $\Psi \left(\theta \right)=\psi_{*},$ namely, $m\left(x\;|\;\psi_{*}\right)=\int_{\Theta }f_{\theta }\left(x\right)\;\Pi \left(d\theta \;|\;\psi_{*}\right),$ and not on the marginal prior $\pi_{\Psi }$ on ψ. We refer to probabilities that depend only on $M(\cdot \;\;|\;\psi_{*})$ as frequentist, for example, coverage probabilities are called confidences, and those that depend on the full prior π as Bayesian confidences. The frequentist label is similar to use of the confidence terminology when dealing with random effects’ models as nuisance parameters have been integrated out.

Suppose now that some other general rule, not necessarily the principle of evidence, is used to determine whether there is evidence in favor of or against $\psi_{*}$ and this leads to the set $D(\psi_{*})\subset X$ as those data sets that do not give evidence in favor of $H_{0}=\left\{\psi_{*}\right\}.$ The rules of potential interest will satisfy $M\left(D\left(\psi_{*}\right)\;|\;\psi_{*}\right)\leq M\left(RB_{\Psi }\left(\psi_{*}\;|\;X\right)\leq 1\;|\;\psi_{*}\right)$ since this implies better performance a priori in terms of identifying when data has evidence in favor of $H_{0}$ via the set $D^{c}\left(\psi_{*}\right)$ than the principle of evidence. For example, $D\left(\psi_{*}\right)=\left\{x:RB_{\Psi }\left(\psi_{*}\;|\;x\right)\leq q\right\}$ for some q<1 satisfies this, but note that a value satisfying $q<RB_{\Psi }\left(\psi_{*}\;|\;x\right)\leq 1$ violates the principle of evidence if it is claimed that there is evidence in favor of $\psi_{*}$. Putting $R\left(\psi_{*}\right)=\left\{x:RB_{\Psi }\left(\psi_{*}\;|\;x\right)\leq 1\right\}$ leads to the following result.

**Theorem** **1.**
*Consider $D(\psi_{*})\subset X$ satisfying $M\left(D\left(\psi_{*}\right)\;|\;\psi_{*}\right)\leq M\left(R\left(\psi_{*}\right)\;|\;\psi_{*}\right).$ (i) The prior probability $M(D\left(\psi_{*}\right))$ is maximized among such rules by $D\left(\psi_{*}\right)=R\left(\psi_{*}\right).$ (ii) If $\Pi_{\Psi }\left(\left\{\psi_{*}\right\}\right)=0,$ then $R(\psi_{*})$ maximizes the prior probability of not obtaining evidence in favor of $\psi_{*}$ when it is false and otherwise maximizes this probability among all rules satisfying $M\left(D\left(\psi_{*}\right)\;|\;\psi_{*}\right)=M\left(R\left(\psi_{*}\right)\;|\;\psi_{*}\right).$*


When $\Pi_{\Psi }\left(\left\{\psi_{*}\right\}\right)\neq 0,$ rules may exist having greater prior probability of not getting evidence in favor of $\psi_{*}$ when it is false, but the price paid for this is the violation of the principle of evidence. In addition, when comparing rules based on their ability to distinguish falsity, it only seems fair that the rules perform the same under the truth. Thus, Theorem 1 is a general optimality result for the principle of evidence applied to hypothesis assessment when considering bias against.

Now, consider C(x)={ψ:x∉D(ψ)}, the set of ψ values for which there is evidence in their favor after observing *x* according to some alternative evidence rule. Since $M\left(\psi_{*}\notin C\left(X\right)\;|\;\psi \right)=M\left(D\left(\psi_{*}\right)\right)\;\left|\;\psi \right),$ then
$$\begin{array}{cc} E_{\Pi_{\Psi }}\left(M\left(\psi \in C\left(X\right)\;\right)\;|\;\psi \right)) & \;=\;1-E_{\Pi_{\Psi }}\left(M\left(\psi \notin C\left(X\right)\;|\;\psi \right)\right)\;=\;1-E_{\Pi_{\Psi }}\left(M\left(D\left(\psi \right)\;|\;\psi \right)\right)\\ \; & \geq 1-E_{\Pi_{\Psi }}\left(M\left(R\left(\psi \right)\;|\;\psi \right)\right)=E_{\Pi_{\Psi }}\left(M\left(\psi \in Pl_{\Psi }\left(X\right)\;\right)\;|\;\psi \right)) \end{array}$$
and so the Bayesian coverage of *C* is at least as large as that of $Pl_{\Psi }$ and thus represents a viable alternative to using $Pl_{\Psi }.$ The following establishes an optimality result for $Pl_{\Psi }$.

**Theorem** **2.**
*(i) The prior probability that the region C doesn’t cover a value $\psi_{*}$ generated from the prior, namely, $E_{\Pi_{\Psi }}\left(M\left(\psi_{*}\notin C\left(X\right)\right)\right),$ is maximized among all regions satisfying*
$$M\left(\psi_{*}\notin C\left(X\right)\;|\;\psi_{*}\right)\leq M\left(\psi_{*}\notin Pl_{\Psi }\left(X\right)\;|\;\psi_{*}\right)$$
*for every $\;\psi_{*},$ by $C=Pl_{\Psi }.$ (ii) If $\Pi_{\Psi }\left(\left\{\psi_{*}\right\}\right)=0$ for all $\psi_{*},$ then $Pl_{\Psi }$ maximizes the prior probability of not covering a false value and otherwise maximizes this probability among all C satisfying $M\left(\psi_{*}\notin C\left(X\right)\;|\;\psi_{*}\right)=M\left(\psi_{*}\notin Pl_{\Psi }\left(X\right)\;|\;\psi_{*}\right)$ for all $\psi_{*}.$*


Again, when $\Pi_{\Psi }\left(\left\{\psi_{*}\right\}\right)\neq 0$, the existence of a region with better properties with respect to not covering false values than $Pl_{\Psi }$ can’t be ruled out, but, when considering such a property, it seems only fair to compare regions with the same coverage probability, and, in that case, $Pl_{\Psi }$ is optimal. Thus, Theorem 2 is also a general optimality result for the principle of evidence applied to estimation when considering bias against. In addition, if there is a value $\psi_{0}=\arg\inf_{\psi }M\left(\psi \in Pl_{\Psi }\left(X\right)\;\right)\;\left|\;\psi \right),$ then $\gamma_{0}=M\left(\psi_{0}\in Pl_{\Psi }\left(X\right)\;\right)\;\left|\;\psi_{0}\right)$ serves as a lower bound on the coverage probabilities, and thus $Pl_{\Psi }$ is a $\gamma_{0}$-confidence region for ψ and this is a pure frequentist $\gamma_{0}$-confidence region when Ψ(θ)=θ. Since $M\left(\psi \in Pl_{\Psi }\left(X\right)\;\right)\;|\;\psi )=1-M\left(\psi \notin Pl_{\Psi }\left(X\right)\;\right)\;\left|\;\psi \right)=1-M\left(R\left(\psi \right)\;|\;\psi \right),$ then Example 3 shows that it is reasonable to expect that such a $\psi_{0}$ exists.

The principle of evidence leads to the following satisfying properties which connect the concept of bias as discussed here with the frequentist concept.

**Theorem** **3.**
*(i) Using the principle of evidence, the prior probability of getting evidence in favor of $\psi_{*}$ when it is true is greater than or equal to the prior probability of getting evidence in favor of $\psi_{*}$ given that $\psi_{*}$ is false. (ii) The prior probability of $Pl_{\Psi }$ covering the true value is always greater than or equal to the prior probability of $Pl_{\Psi }$ covering a false value.*


The properties stated in Theorem 3 are similar to a property called unbiasedness for frequentist procedures. For example, a test is unbiased if the probability of rejecting a null is always larger when it is false than when it is true and a confidence region is unbiased if the probability of covering the true value is always greater than the probability of covering a false value. While the inferences discussed here are “unbiased” in this generalized sense, they could still be biased against or in favor in the sense of this paper, as it is the amount of data that controls this.

Now, consider bias in favor and suppose there is an alternative characterization of evidence that leads to the region $E(\psi_{*})$ consisting of all data sets that do not lead to evidence against $\psi_{*}.$ Putting $A\left(\psi_{*}\right)=\left\{x:RB_{\Psi }\left(\psi_{*}\;|\;x\right)\geq 1\right\},$ we restrict attention to regions satisfying $M\left(E\left(\psi_{*}\right)\;|\;\psi_{*}\right)\geq M\left(A\left(\psi_{*}\right)\;|\;\psi_{*}\right).$ Using (Equation 4) to measure bias in favor leads to the following results.

**Theorem** **4.**
*(i) The prior probability $M(E\left(\psi_{*}\right))$ is minimized among all $E(\psi_{*})\subset X$ satisfying $M\left(E\left(\psi_{*}\right)\;|\;\psi_{*}\right)\geq M\left(A\left(\psi_{*}\right)\;|\;\psi_{*}\right)$ by $E\left(\psi_{*}\right)=A\left(\psi_{*}\right).$ (ii) If $\Pi_{\Psi }\left(\left\{\psi_{*}\right\}\right)=0,$ then the set $A(\psi_{*})$ minimizes the prior probability of not obtaining evidence against $\psi_{*}$ when it is false and otherwise minimizes this probability among all rules satisfying $M\left(E\left(\psi_{*}\right)\;|\;\psi_{*}\right)=M\left(A\left(\psi_{*}\right)\;|\;\psi_{*}\right).$*


**Theorem** **5.**
*(i) The prior probability region C covers a value $\psi_{*}$ generated from the prior, namely, $E_{\Pi_{\Psi }}\left(M\left(\psi_{*}\in C\left(X\right)\right)\right),$ is minimized among all regions satisfying $M\left(\psi_{*}\in C\left(X\right)\;|\;\psi_{*}\right)\geq M\left(\psi_{*}\in Pl_{\Psi }\left(X\right)\;|\;\psi_{*}\right)$ for every $\;\psi_{*},$ by $C=Pl_{\Psi }.$ (ii) If $\Pi_{\Psi }\left(\left\{\psi_{*}\right\}\right)=0$ for all $\psi_{*},$ then $Pl_{\Psi }$ minimizes the prior probability of covering a false value and otherwise minimizes this probability among all rules satisfying $M\left(\psi_{*}\in C\left(X\right)\;|\;\psi_{*}\right)=M\left(\psi_{*}\in Pl_{\Psi }\left(X\right)\;|\;\psi_{*}\right)$ for all $\psi_{*}.$*


Thus, Theorems 4 and 5 are optimality results for the principle of evidence when considering bias in favor.

Clearly, the bias against $H_{0}$ is playing a role similar to size in frequentist statistics and the bias in favor is playing a role similar to power. A study that found evidence against $H_{0},$ but had a high bias against, or a study that found evidence in favor of $H_{0}$ but had a high bias in favor, could not be considered to be of high quality. Similarly, a study concerned with estimating a quantity of interest could not be considered of high quality if there is high bias against or in favor. There are some circumstances, however, where some bias is perhaps not an issue. For example, in a situation where sparsity is to be expected, then, allowing for high bias in favor of certain hypotheses accompanied by low bias against, may be tolerable, although this does reduce the reliability of any hypotheses where evidence is found in favor.

The concept of a *severe test* is introduced in [32], and this has a similar motivation to measuring bias. This is described now with some small modifications that allow for a more general discussion than the special situations used in the reference. Suppose d(x) is the test statistic for an test of size α so that $H_{0}:\Psi \left(\theta \right)=\psi_{0}$ is rejected when $d\left(x\right)>c_{\alpha }$ and accepted otherwise. A deviation $\gamma^{*}$ that is *substantively important* is specified. When the test leads to the acceptance of $H_{0}$, the severity of the test is assessed by the *attained power*
$P_{\theta }\left(d\left(X\right)>d\left(x\right)\;|\;x\right)\;$for θ values satisfying $d_{\Psi }\left(\psi_{0},\Psi \left(\theta \right)\right)\geq \gamma^{*},$ where $d_{\Psi }$ is a distance measure on Ψ. To get a single number for the severity measure, it makes sense to use $\inf_{\left\{\theta :d_{\Psi }\left(\psi_{0},\Psi \left(\theta \right)\right)=\gamma^{*}\right\}}P_{\theta }\left(d\left(X\right)>d\left(x\right)\;|\;x\right)$ as generally $P_{\theta }\left(d\left(X\right)>d\left(x\right)\;|\;x\right)$ will increase as $d_{\Psi }\left(\psi_{0},\Psi \left(\theta \right)\right)$ increases. The hypothesis $H_{0}$ is accepted with *high severity* when the attained power is high. The motivation for adding this measure of the test is that it claimed that it is incorrect to simply accept $H_{0}$ when $d\left(x\right)\leq c_{\alpha }$ unless the probability of obtaining a value of the test statistic as least as large as that observed is high when the hypothesis is meaningfully false. When $H_{0}$ is rejected, then the severity of the test is measured by $P_{\theta }\left(d\left(X\right)\leq d\left(x\right)\;|\;x\right)\;$for θ values satisfying $d_{\Psi }\left(\psi_{0},\Psi \left(\theta \right)\right)<\gamma^{*}$ and, to obtain a single number one could use $\sup_{\left\{\theta :d_{\Psi }\left(\psi_{0},\Psi \left(\theta \right)\right)\leq \gamma^{*}\right\}}P_{\theta }\left(d\left(X\right)\leq d\left(x\right)\;|\;x\right).$ It is then required that this probability be small to claim a rejection with high severity.

The use of the $\gamma^{*}$ quantity seems identical to the difference that matters δ and we agree that this is an essential aspect of a statistical analysis. In hypothesis assessment, this guards against “the large *n* problem” where large sample sizes will detect deviations from $H_{0}$ that are not practically meaningful. There are, however, numerous differences with the discussion of bias here. The severity approach is expressed within the context where either $H_{0}$ or $H_{0}^{c}$ is accepted and the relative belief approach is more general than this binary classification. The testing approach suffers from the lack of a clear choice of α to determine the cut-off, and this is not the case for the principle of evidence. The bias measures are frequentist performance characteristics, albeit somewhat dependent on the prior, but the measures of severity are conditional on the observed *x* leaving one wondering about their frequentist performance characteristics, see [33] for more discussion on this point. The assessment of $H_{0}$ via relative belief is based on the observed data and datasets not observed are irrelevant, at least for the expression of the evidence. The relevance of unobserved data are for us better addressed a priori where such considerations lead to an assessment of the merits of the study, but these play no role in the actual inferences. The major difference is that a proper prior is required here as this leads to a characterization of evidence via the principle of evidence.

## 5. Examples

A number of examples are now considered.

**Example** **4.**
*Binomial proportion.*


Suppose $x=(x_{1},\ldots ,x_{n})$ is a sample from the Bernoulli(θ) with θ∈[0,1] unknown so $n\bar{x}∼$ binomial(n,θ) and interest is in θ. For the prior, let θ∼ beta$\left(\alpha_{0},\beta_{0}\right)$ where the hyperparameters are elicited as in, for example [34], so $\theta \;|\;n\bar{x}∼$ beta$(\alpha_{0}+n\bar{x},\beta_{0}+n\left(1-\bar{x}\right)).$ Then,
$$RB\left(\theta \;|\;n\bar{x}\right)=\frac{\Gamma (\alpha_{0}+\beta_{0}+n)}{\Gamma \left(\alpha_{0}+n\bar{x}\right)\Gamma \left(\beta_{0}+n\left(1-\bar{x}\right)\right)}\frac{\Gamma \left(\alpha_{0}\right)\Gamma \left(\beta_{0}\right)}{\Gamma (\alpha_{0}+\beta_{0})}\theta^{n\bar{x}}{\left(1-\theta \right)}^{n(1-\bar{x})}$$
is unimodal with mode at $\bar{x},$ so Pl(x) is an interval containing $\bar{x}.$ Note that M(·|θ) is the binomial(n,θ) probability measure and the bias against θ is given by $M(RB\left(\theta \;|\;n\bar{x}\right)\leq 1\;|\;\theta )$ while the bias in favor of θ, using (Equation 5), is given by $\max M(RB\left(\theta \;|\;n\bar{x}\right)\geq 1\;|\;\theta \pm \delta )$ for θ∈[δ,1−δ].

Consider first the prior given by $\left(\alpha_{0},\beta_{0}\right)=\left(1,1\right).$
Figure 4a gives the plots of the bias against for n=10 (max. = 0.21, average = 0.11), n=50 (max.= 0.07, average = 0.05) and n=100 (max. = 0.05, average = 0.03). Therefore, when n=10, then Pl(x) is a 0.79-confidence interval for θ; when n=50, it is a 0.93-confidence interval for θ and, when n=100, it is a 0.95-confidence interval for θ. For the informative prior given by $\left(\alpha_{0},\beta_{0}\right)=\left(5,5\right)$, Figure 4b gives the plots of the bias against for n=10 (max. = 0.36, average = 0.21), n=50 (max. = 0.16, average = 0.10) and n=100 (max. = 0.11, average = 0.07). Thus, when n=10, then Pl(x) is a 0.64-confidence interval for θ, when n=50, it is a 0.84-confidence interval for θ and, when n=100, it is a 0.93-confidence interval for θ. One feature immediately stands out, namely, when using a more informative prior the bias against increases. As previously explained, this phenomenon occurs because when the prior probability of θ is small, it is much easier to obtain evidence in favor than when the prior probability of θ is large.

Now, consider bias in favor using (Equation 11). When $\left(\alpha_{0},\beta_{0}\right)=\left(1,1\right)$ and δ=0.1,
Figure 5a gives the plots of the bias in favor for n=10 (max. = 1.00, average = 0.84), n=50 (max. = 0.72, average = 0.51) and n=100 (max. = 0.50, average = 0.35). Therefore, when n=10, the maximum probability that Pl(x) contains a false value at least δ away from the true value is 1, when n=50 this probability is 0.72 and, when n=100, it is a 0.50. When $\left(\alpha_{0},\beta_{0}\right)=\left(5,5\right),$
Figure 5b gives the plots of the bias in favor for n=10 (max. = 1.00, average = 0.68), for n=50 (max. = 1.00, average = 0.71) and for n=100 (max. = 1.00, average = 0.49). Thus, in this case, the maximum probability that Pl(x) contains a false value at least δ away from the true value is always 1, but, when averaged with respect to the prior, the values are considerably less. It is necessary to either increase *n* or δ to decrease bias in favor. For example, with $\left(\alpha_{0},\beta_{0}\right)=\left(5,5\right),$δ=0.1 and n=400, the maximum bias in favor is 0.02 and the average bias in favor is 0.02 and, when n=600, these quantities equal 0 to two decimals. When δ=0.2 and n=50, the maximum bias in favor is 0.29 and the average bias in favor is 0.11 and, when n=100, the maximum bias in favor is 0.01 and the average bias in favor is 0.01.

Another interesting case is when the prior is taken to be Jeffreys prior which in this case is the beta(1/2,1/2) distribution. This reference prior, see [35], is proper and thus can be used with the principle of evidence. The prior does represent somewhat extreme beliefs, however, as 28.7% of the beliefs are that θ∈(0,0.05)∪(0.95,1). The corresponding biases against are for n=10 (max. = 0.24, average = 0.07), n=50 (max. = 0.09, average = 0.03) and n=100 (max. = 0.07, average = 0.02). The biases in favor are, using (Equation 11) with δ=0.1, for n=10 (max. = 1.00, average = 0.73), n=50 (max. = 0.72, average = 0.59) and n=100 (max. = 0.54, average = 0.41). Although the plots of the bias functions can be seen to be quite different than those for the beta(1,1) prior, the summary values presented are very similar. The beta(1/2,1/2) prior does a bit better with respect to bias against but a bit worse with respect to bias in favor. This reinforces the point that the biases do not serve as a basis for the choice of the prior.

The strange oscillatory nature of the plots for the binomial is difficult to understand but is a common feature with such calculations. For example, Ref. [36] studies the coverage probabilities for various confidence intervals for the binomial, and the following comment is made “The oscillation in the coverage probability is caused by the discreteness of the binomial distribution, more precisely, the lattice structure of the binomial distribution”, which still doesn’t fully explain the phenomenon.

**Example** **5.**
*Location-scale normal quantiles.*


Suppose $x=(x_{1},\ldots ,x_{n})$ is a sample from $N(\mu ,\sigma^{2})$ with $\left(\mu ,\sigma^{2}\right)\in R^{1}\times \left(0,\infty \right)$ unknown with prior $\mu \;|\;\sigma^{2}∼N\left(\mu_{0},\tau_{0}^{2}\sigma^{2}\right),\sigma^{-2}∼\;$gamma${}_{\mathrm{rate}}\left(\alpha_{0},\beta_{0}\right)$. The hyperparameters $\left(\mu_{0},\tau_{0}^{2},\alpha_{0},\beta_{0}\right)$ can be obtained via an elicitation as, for example, discussed in Evans and Tomal (2018) for the more general regression model. This example is easily generalized to the regression context. A MSS is $T\left(x\right)=(\bar{x},||x-\bar{x}1|{|}^{2}),$ where $1={\left(1,\ldots ,1\right)}^{\prime }$, with the posterior distribution given by $\mu \;|\;\sigma^{2},T\left(x\right)∼N\left(\mu_{0x},{\left(n+1/\tau_{0}^{2}\right)}^{-1}\sigma^{2}\right),\sigma^{-2}\;|\;T\left(x\right)∼\;$gamma${}_{rate}\left(\alpha_{0}+n/2,\beta_{0x}\right),$ where $\mu_{0x}={\left(n+1/\tau_{0}^{2}\right)}^{-1}\left(n\bar{x}+\mu_{0}/\tau_{0}^{2}\right)$ and
$$\beta_{0x}=\beta_{0}+||x-\bar{x}1{\left|\right|}^{2}/2+n{\left(\bar{x}-\mu_{0}\right)}^{2}/2\left(n\tau_{0}^{2}+1\right).$$

Suppose interest is in the γ-th quantile $\psi =\Psi \left(\mu ,\sigma^{2}\right)=\mu +\sigma z_{\gamma },$ where $z_{\gamma }=\Phi^{-1}\left(\gamma \right).$ To determine the bias for or against ψ, we need the prior and posterior densities of ψ for which there is not a closed form. It is easy, however, to work with the discretized ψ by simply generating from the prior and posterior of $(\mu ,\sigma^{2}),$ estimate the contents of the relevant intervals and then approximate the relative belief ratio using these. Thus, we are essentially approximating the densities by density histograms here, although alternative density estimates could be used. A natural approach to the discretization is to base it on the prior mean $E\left(\psi \right)=\mu_{0}+\beta_{0}^{1/2}\left(\Gamma \left(\alpha_{0}-1/2\right)/\Gamma \left(\alpha_{0}\right)\right)z_{\gamma }$ and variance $Var\left(\psi \right)=E\left(\psi^{2}\right)-{\left(E\left(\psi \right)\right)}^{2}$ where $E\left(\psi^{2}\right)=\left(z_{\gamma }^{2}+\tau_{0}^{2}\right)\beta_{0}/\left(\alpha_{0}-1\right).$ Thus, for a given δ, we discretize using 2k+1 intervals (E(ψ)+iδ,E(ψ)+(i+1)δ] where k=cSD(ψ)/δ and *c* is chosen so that the collection of intervals covers the effective support of ψ which is easily assessed as part of the simulation. For example, with the prior given by hyperparameters $\mu_{0}=0,\tau_{0}^{2}=1,\alpha_{0}=2,\beta_{0}=1$ and γ=0.5,δ=0.1,c=5, then k=50 and, on generating $10^{5}$ values from the prior, these intervals contained 99,699 of the values and with c=6, then k=60, and these intervals contained 99,901 of the generated values. Similar results are obtained for more extreme quantiles because the intervals shift.

For the bias against for estimation, the value of $M(RB_{\Psi }\left(\psi \;|\;X\right)\leq 1\;|\;\psi )$ is needed for a range of ψ values. For this, we need to generate from the conditional prior distribution of *T* given $\Psi (\mu ,\sigma^{2})=\psi$, and an algorithm for generating from the conditional prior of $\left(\mu ,\sigma^{2}\right)$ given ψ is needed. Putting $\nu =1/\sigma^{2},$ the transformation $\left(\mu ,\nu \right)\to \left(\psi ,\nu \right)=\left(\mu +\nu^{-1/2}z_{\gamma },\nu \right)$ has Jacobian equal to 1, so the conditional prior distribution of ν|ψ has density proportional to $\nu^{\alpha_{0}-1/2}\exp\left\{-\beta_{0}\nu \right\}\exp\left\{-\nu {\left(\psi -\mu_{0}-\nu^{-1/2}z_{\gamma }\right)}^{2}/2\tau_{0}^{2}\right\}.$ The following gives a rejection algorithm for generating from this distribution:generate ν∼ gamma$(\alpha_{0}+1/2,\beta_{0}),$generate u∼ unif(0,1) independent of ν,if $u\leq \exp\{-\nu {\left(\psi -\mu_{0}-\nu^{-1/2}z_{\gamma }\right)}^{2}/2\tau_{0}^{2}\}$ return ν, else go to 1.

As ψ moves away from the prior expected value E(ψ), this algorithm becomes less efficient, but, even when the expected number of iterations is 86 (when γ=0.95,ψ=12), generating a sample of $10^{4}$ is almost instantaneous. Figure 6 is a plot of the conditional prior of ν given that ψ=2. After generating ν, then generate $||x-\bar{x}1{\left|\right|}^{2}∼$$\nu^{-1}$chi-squared(n−1) and $\bar{x}∼N\left(\psi -\nu^{-1/2}z_{\gamma },\nu^{-1}/n\right)$ to complete the generation of a value from $M_{T}\left(\cdot \;|\;\psi \right).$

The bias against as a function of $\psi =\mu +\sigma z_{0.95},$ has maximum value 0.151 when n=10 and so $Pl_{\Psi }\left(x\right)$ is a 0.849-confidence region for ψ while the average bias against is 0.104 implying that the Bayesian coverage is 0.896.
Table 7 gives the coverages for other values of *n* as well. Figure 7 is a plot of the bias in favor as a function of ψ with δ=±0.5 and n=10. The jitter in the right tail is a result of Monte Carlo sampling error, but this error is not of significance as bias measurements are not required to be known to high accuracy. The average bias in favor is 0.629. When n=50, the average bias in favor is 0.335.

The case γ=0.50, so $\psi =\Psi (\mu ,\sigma^{2})=\mu$ is also of interest. For n=10, then $Pl_{\Psi }\left(x\right)$ has 0.878 frequentist coverage and 0.926 Bayesian coverage; when n=20, the coverages are 0.916 and 0.952 while, when n=50, the coverages are 0.950 and 0.973. When n=10,δ=0.5, the average bias in favor is 0.619; when n=20, this is 0.4206 and, for n=100, the average bias in favor is 0.091.

**Example** **6.**
*Normal Regression—Prediction.*


Prediction problems have some unique aspects when compared to inferences about parameters. To see this, consider first the location normal model of Example 3, and the problem is to make an inference about a future value $y∼N(\mu ,\sigma_{0}^{2}).$ The prior predictive distribution is $y∼N(\mu_{0},\tau_{0}^{2}+\sigma_{0}^{2})$ and the posterior predictive is $y∼N(\mu_{x},\sigma_{n}^{2}+\sigma_{0}^{2})$ where $\mu_{x}=\sigma_{n}^{2}\left(n\bar{x}/\sigma_{0}^{2}+\mu_{0}/\tau_{0}^{2}\right),\sigma_{n}^{2}={\left(n/\sigma_{0}^{2}+1/\tau_{0}^{2}\right)}^{-1}$ so
$$RB\left(y\;|\;\bar{x}\right)={\left(\frac{\tau_{0}^{2}+\sigma_{0}^{2}}{\sigma_{n}^{2}+\sigma_{0}^{2}}\right)}^{1/2}\exp\left\{-\frac{1}{2}\left[\frac{{\left(y-\mu_{x}\right)}^{2}}{\sigma_{n}^{2}+\sigma_{0}^{2}}-\frac{{\left(y-\mu_{0}\right)}^{2}}{\tau_{0}^{2}+\sigma_{0}^{2}}\right]\right\}.$$

For a given *y*, the bias against is $M(RB\left(y\;|\;\bar{x}\right)\leq 1\;|\;y)$ and, for this, we need the conditional prior predictive of $\bar{x}\;|\;y.$ The joint prior predictive is $\left(\bar{x},\;y\right)∼N_{2}\left(\mu_{0}1_{2},\Sigma_{0}\right)$, where
$$\Sigma_{0}=\left(\begin{array}{cc} \tau_{0}^{2}+\sigma_{0}^{2}/n & \tau_{0}^{2}\\ \tau_{0}^{2} & \tau_{0}^{2}+\sigma_{0}^{2} \end{array}\right)$$
and so $\bar{x}\;|\;y∼N\left(\mu_{0}+\tau_{0}^{2}\left(y-\mu_{0}\right)/\left(\tau_{0}^{2}+\sigma_{0}^{2}\right),\sigma_{0}^{2}\left(\tau_{0}^{2}/\left(\tau_{0}^{2}+\sigma_{0}^{2}\right)+1/n\right)\right).$ From this, we see that, as n→∞, the conditional prior distribution of $\mu_{x}\;|\;y$ converges to the
$$N\left(\mu_{0}+\tau_{0}^{2}\left(y-\mu_{0}\right)/\left(\tau_{0}^{2}+\sigma_{0}^{2}\right),\sigma_{0}^{2}\tau_{0}^{2}/\left(\tau_{0}^{2}+\sigma_{0}^{2}\right)\right)$$
distribution. Thus, with Z∼N(0,1), $r=\tau_{0}^{2}/\sigma_{0}^{2}$, and
$$d\left(\left(y-\mu_{0}\right)/\sigma_{0},r\right)=\left(1+1/r\right)\log\left(1+r\right)+r^{-1}{\left(y-\mu_{0}\right)}^{2}/\sigma_{0}^{2}),$$
then
$$M\left(RB\left(y\;|\;\bar{x}\right)\leq 1\;|\;y\right)\to 1-P\left(Z\in \left[r^{-1/2}{\left(1+r\right)}^{-1/2}\left(y-\mu_{0}\right)/\sigma_{0}\pm d^{1/2}\left(\left(y-\mu_{0}\right)/\sigma_{0},r\right)\right]\right)$$
as n→∞. Thus, the bias against does not go to 0 as n→∞, and there is a limiting lower bound to the prior probability that evidence in favor of a specific *y* will not be obtained. This baseline is dependent on both $\left(y-\mu_{0}\right)/\sigma_{0}$ and *r*. As $r=\tau_{0}^{2}/\sigma_{0}^{2}\to \infty$, this baseline bias against goes to 0 and so it is necessary to ensure that the prior variance is not too small. Table 8 gives some values for the bias against, and it is seen that, if $\tau_{0}^{2}/\sigma_{0}^{2}$ is too small, then there is substantial bias against even when *y* is a reasonable value from the distribution. When $\tau_{0}^{2}/\sigma_{0}^{2}=1,\left(y-\mu_{0}\right)/\sigma_{0}=0$ and n=10, the bias against is computed to be 0.248, which is quite close to the baseline, so increasing sample size will not reduce bias against by much and similar results are obtained for the other cases.

Now consider bias in favor of *y*, namely, $M(RB\left(y\;|\;\bar{x}\right)\geq 1\;|\;y\pm \delta )$ for some choice of δ. False values for *y* correspond to values in the tails so we consider, for example, y+δ as a value in the central region of the prior and then a large value of δ puts *y* in the tails. Again, the bias in favor has a baseline value as n→∞. A similar argument leads to the bias in favor of *y* satisfying
$$\begin{array}{cc} \; & M(RB\left(y\;|\;\bar{x}\right)\geq 1\;|\;y\pm \delta )\to \\ \; & P\left(Z\in \left[r^{-1/2}{\left(1+r\right)}^{-1/2}\left(\frac{y-\mu_{0}}{\sigma_{0}}\pm r\frac{\delta }{\sigma_{0}}\right)\pm d^{1/2}\left(\frac{y-\mu_{0}}{\sigma_{0}},r\right)\right]\right). \end{array}$$

Figure 8 is a plot of $\sup M(RB\left(y\;|\;\bar{x}\right)\geq 1\;|\;y\pm \delta ).$ Thus, the bias in favor is low for central values of *y*, but, once again, there is a trade-off as when *r* increases the bias in favor goes to 1.

Prediction plays a bigger role in regression problems, but we can expect the same issues to apply as in the location problem. Suppose $y∼N_{n}\left(X\beta ,\sigma^{2}I\right)$, where $X\in R^{n\times k}$ is of rank k,
$\left(\beta ,\sigma^{2}\right)\in R^{k}\times \left(0,\infty \right)$ is unknown, our interest is in predicting a future value $y_{new}∼N\left(w^{t}\beta ,\sigma^{2}\right)$ for some fixed known *w* and, putting $\nu =1/\sigma^{2},$ the conjugate prior $\beta \;|\;\nu ∼N_{k}\left(\beta_{0},\nu^{-1}\Sigma_{0}\right)\;\nu ∼\;$gamma${}_{\mathrm{rate}}\left(\alpha_{0},\eta_{0}\right)$ is used. Specifying the hyperparameters $\left(\beta_{0},\Sigma_{0},\alpha_{0},\eta_{0}\right)$ can be carried out using elicitation as discussed in [37].

For the bias calculations, it is necessary to generate values of the MSS $\left(b,s^{2}\right)=({\left(X^{t}X\right)}^{-1}X^{t}y,||y-{Xb||}^{2})$ from the conditional prior predictive $M(\cdot \;|\;y_{new}).$ This is accomplished by generating from the conditional prior of $\left(\beta ,\nu \right)\;|\;y_{new}$ and then generating $b∼N_{k}\left(\beta ,\nu^{-1}{\left(X^{t}X\right)}^{-1}\right)$ independent of $s^{2}∼\nu^{-1}\;$chi-squared(n−k). The conditional prior of $\left(\beta ,\nu \right)\;|\;y_{new}$ is proportional to
$$\begin{array}{cc} \; & \nu^{\alpha_{0}-1/2}\exp\left\{-\eta_{0}\left(y_{new}\right)\nu \right\}\times \\ \; & \nu^{k/2}\exp\left\{-\frac{\nu }{2}{\left(\beta -{\left(\Sigma_{0}^{-1}+ww^{t}\right)}^{-1}\left(\Sigma_{0}^{-1}\beta_{0}+y_{new}w\right)\right)}^{t}\left(\Sigma_{0}^{-1}+ww^{t}\right)\left(\cdot \right)\right\} \end{array}$$
where
$$\begin{array}{cc} {\left(\Sigma_{0}^{-1}+ww^{t}\right)}^{-1} & =\Sigma_{0}-{\left(1+w^{t}\Sigma_{0}w\right)}^{-1}\Sigma_{0}ww^{t}\Sigma_{0},\eta_{0}\left(y_{new}\right)\\ \; & =\eta_{0}+{\left(1+w^{t}\Sigma_{0}w\right)}^{-1}{\left(w^{t}\beta -y_{new}\right)}^{2}/2. \end{array}$$

Thus, generating $\left(\beta ,\nu \right)\;|\;y_{new}$ is accomplished via ν∼gamma${}_{\mathrm{rate}}\left(\alpha_{0}+1/2,\eta_{0}\left(y_{new}\right)\right)$,
$$\beta \;|\;\nu ∼N_{k}\left(\left(I-\frac{\Sigma_{0}ww^{t}}{1+w^{t}\Sigma_{0}w}\right)\left(\beta_{0}+y_{new}\Sigma_{0}w\right),\nu^{-1}\left(\Sigma_{0}-\frac{\Sigma_{0}ww^{t}\Sigma_{0}}{1+w^{t}\Sigma_{0}w}\right)\right).$$

For each generated $\left(b,s^{2}\right)$, it is necessary to compute the relative belief ratio $RB(y_{new}\;|\;b,s^{2})$ and determine if it is less than or equal to 1. There are closed forms for the prior and conditional densities of $y_{new}$ since $y_{new}∼w^{t}\beta_{0}+{\left\{\eta_{0}\left(1+w^{t}\Sigma_{0}w\right)/\alpha_{0}\right\}}^{1/2}t_{2\alpha_{0}},y_{new}\;|\;\left(b,s^{2}\right)∼w^{t}\beta_{0}\left(b,s^{2}\right)+{\left\{\eta_{0}\left(b,s^{2}\right)\left(1+w^{t}{\left(\Sigma_{0}^{-1}+X^{t}X\right)}^{-1}w\right)/\left(\alpha_{0}+n/2\right)\right\}}^{1/2}t_{2\alpha_{0}+n}$ where $t_{\lambda }$ denotes a Student(λ) random variable and $\beta_{0}\left(b,s^{2}\right)={\left(\Sigma_{0}^{-1}+X^{t}X\right)}^{-1}\left(\Sigma_{0}^{-1}\beta_{0}+X^{t}Xb\right),\eta_{0}\left(b,s^{2}\right)=\eta_{0}+[s^{2}+{\left||Xb|\right|}^{2}+\left|\right|\Sigma_{0}^{-1}\beta_{0}{\left|\right|}^{2}-\beta_{0}{\left(b,s^{2}\right)}^{t}\left(\Sigma_{0}^{-1}+X^{t}X\right)\beta_{0}\left(b,s^{2}\right)]/2.$ These results permit the calculation of the biases as in the location problem.

## 6. Conclusions

There are several conclusions that can be drawn from the discussion here. First, it is necessary to take bias into account when considering Bayesian procedures and currently this is generally not being done. Depending on the purpose of the study, some values concerning both bias against and bias in favor need to be quoted as these are figures of merit for the study. The approach to Bayesian inferences via a characterization of evidence makes this relatively straight-forward conceptually. Second, frequentism can play a role in the approach to Bayesian statistical reasoning via relative belief, not through the inferences, but rather through determining the biases and then controlling these through the amount of data collected. Overall, this makes sense because, before the data are seen, it is natural to be concerned about what inferences can be reliably drawn. Once the data are observed, however, it is the evidence in this data set that matters and not the evidence in the data sets not seen. Still, if we ignore the latter, it may be that the existence of bias makes the inferences drawn of very low quality. Third, the results concerning the standard *p*-value in Example 3 can be seen to apply quite generally, and this makes any discussion about how to characterize and measure evidence of considerable importance. The principle of evidence makes a substantial contribution in this regard as was shown in a variety of results. The major purpose of this paper, however, is to deal with a key criticism of Bayesian methodology, namely that inferences can be biased because of their dependence on the subjective beliefs of the analyst. This criticism is accepted, but we also assert that this can be dealt with in a logical and scientific fashion as has been demonstrated in this paper.

## Figures and Tables

**Figure 1 entropy-23-00190-f001:**
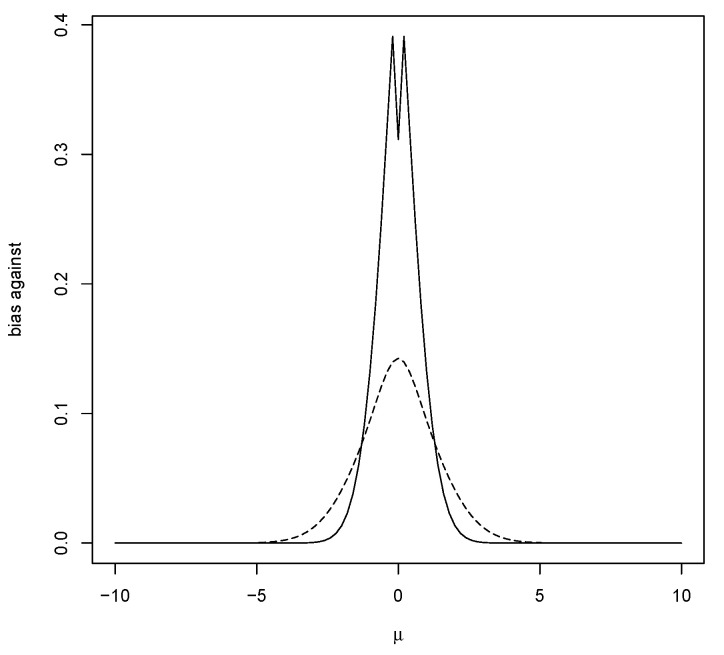
Plot of bias against $H_{0}=\left\{\mu \right\}$ with a N(0,1) prior (- - -) and a N(0,0.01) prior (—) with $n=5,\sigma_{0}=1.$

**Figure 2 entropy-23-00190-f002:**
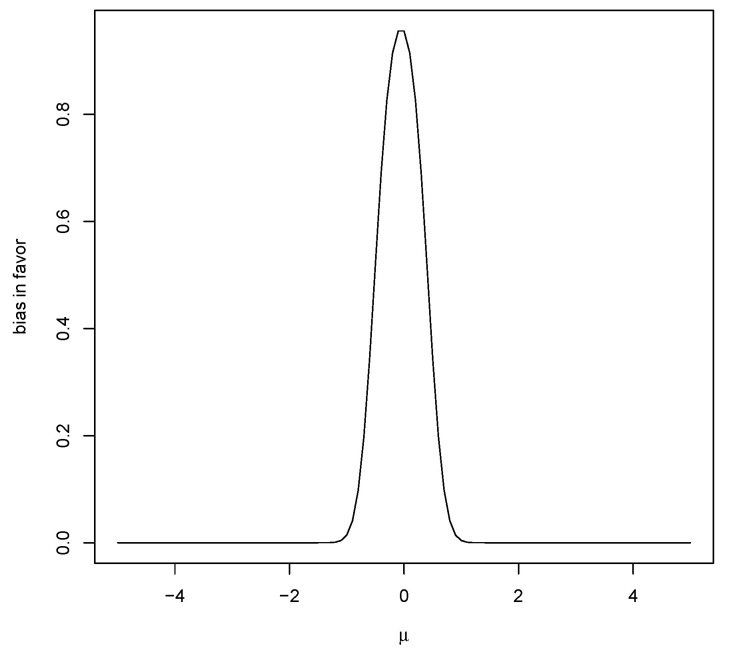
Plot of M(RB(0|X)≥1|μ) when $n=20,\mu_{0}=1,\tau_{0}=1,\sigma_{0}=1.$

**Figure 3 entropy-23-00190-f003:**
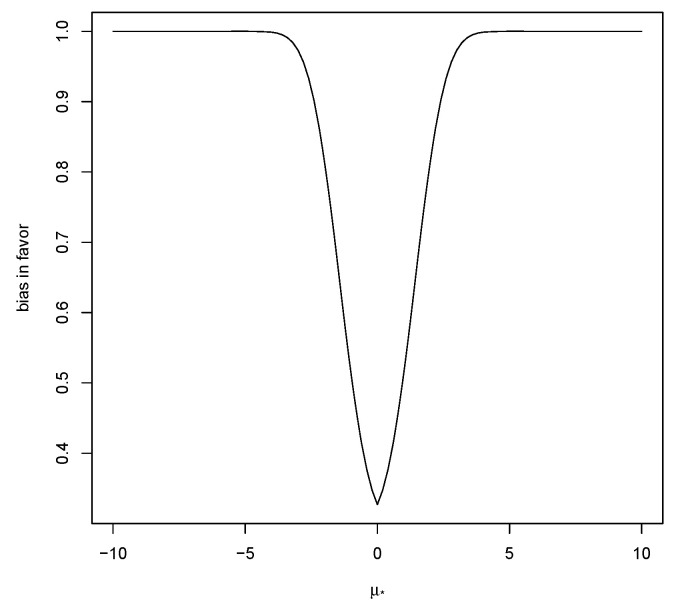
Bias in favor of μ maximized over μ±δ based on a N(0,1) prior and $\sigma_{0}=1,n=20,\delta =0.5.$

**Figure 4 entropy-23-00190-f004:**
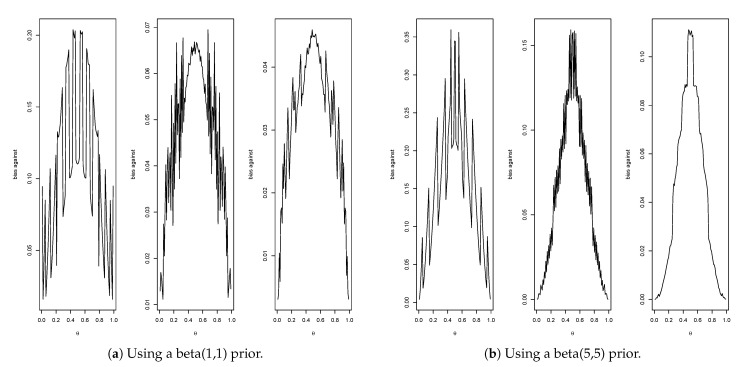
Plots of bias against at θ for n=10,50,100 in Example 4.

**Figure 5 entropy-23-00190-f005:**
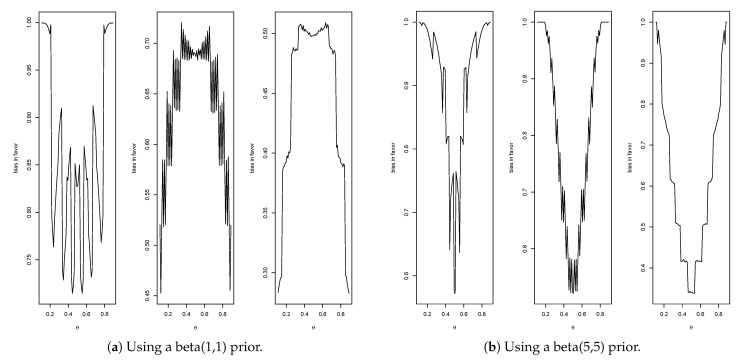
The bias in favor at θ for n=10,50,100 with δ=0.1 in Example 4.

**Figure 6 entropy-23-00190-f006:**
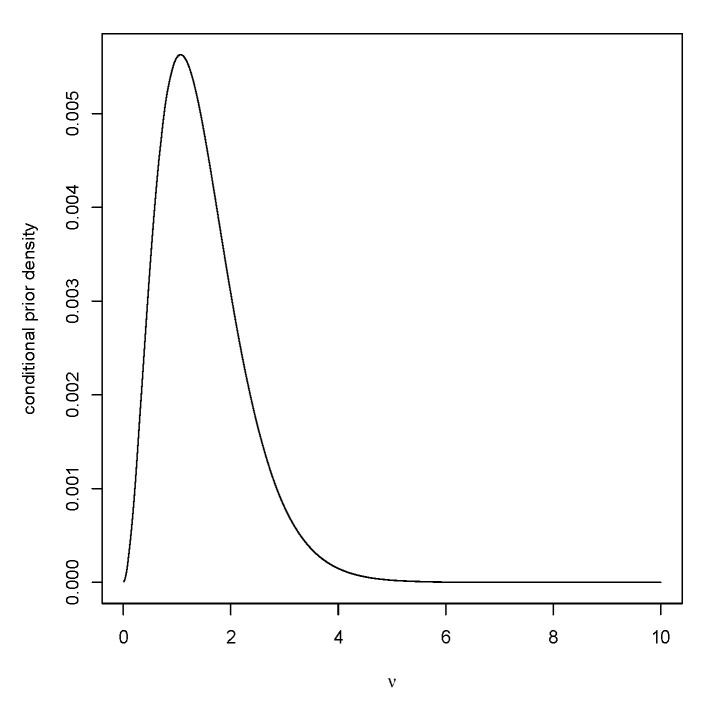
Conditional prior density of $\nu =1/\sigma^{2}$ given ψ=2 when γ=0.95 and $\mu_{0}=0,\tau_{0}^{2}=1$, $\alpha_{0}=2,\beta_{0}=1$ in Example 5.

**Figure 7 entropy-23-00190-f007:**
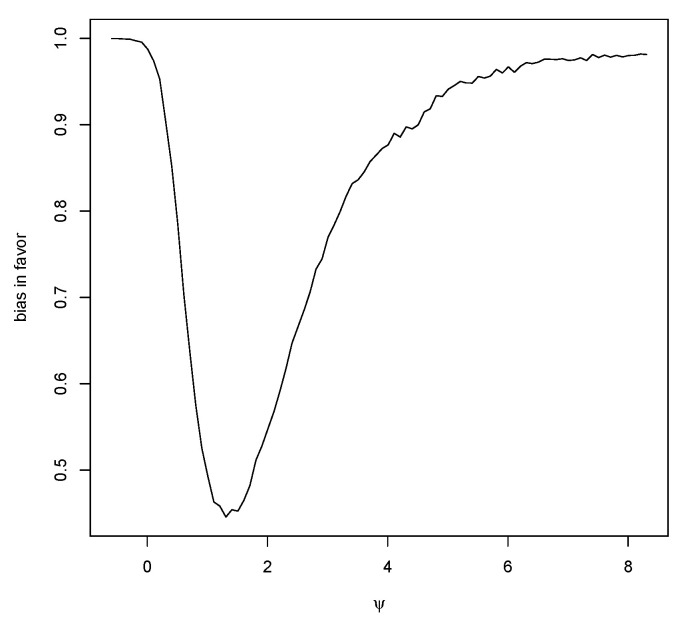
The bias in favor as a function of ψ when γ=0.95,n=10,δ=0.5 and using a prior with hyperparameters $\mu_{0}=0,\tau_{0}^{2}=1,\alpha_{0}=2,\beta_{0}=1$ in Example 5.

**Figure 8 entropy-23-00190-f008:**
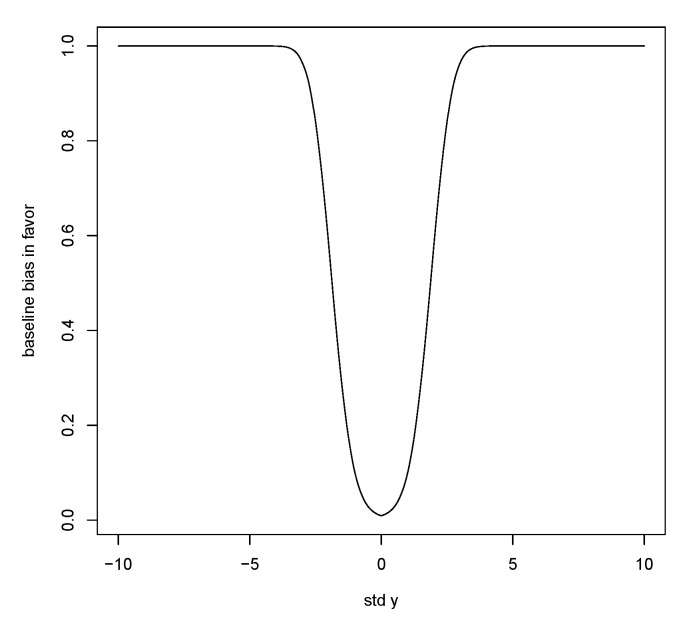
Plot of the baseline bias in favor for values of $\left(y-\mu_{0}\right)/\sigma_{0}$ when $\tau_{0}^{2}/\sigma_{0}^{2}=1$ when δ=5 in Example 6.

**Table 1 entropy-23-00190-t001:** Probabilities and relative belief ratios for $H_{0}$ in Example 1.

	$P(H_{0})$	$P(H_{0}\;|\;C_{k})$		$\mathrm{RB}(H_{0}\;|\;C_{k})$
m=2	0.0045	k=0 k=1 k=2	0.0049 0.0024 0.0008	1.0824 0.5412 0.1804
m=5	0.0399	k=0 k=1 k=2	0.0483 0.0259 0.0093	1.2089 0.6487 0.2317
m=10	0.1431	k=0 k=1 k=2	0.1994 0.1254 0.0522	1.3934 0.8765 0.3652
m=20	0.3481	k=0 k=1 k=2	0.3487 0.4597 0.3831	1.0018 1.3205 1.1004
m=25	0.3890	k=0 k=1 k=2	0.0171 0.2051 0.8547	0.0439 0.5274 2.1974
m=26	0.3902	k=0 k=1 k=2	0.0000 0.0000 1.0000	0.0000 0.0000 2.5630

**Table 2 entropy-23-00190-t002:** Bias against (Equation 3) the hypothesis $H_{0}=\left\{0\right\}$ with a $N(\mu_{0},\tau_{0}^{2})$ prior for different sample sizes *n* with $\sigma_{0}=1$.

*n*	$\mu_{0}=1,\tau_{0}=1$	$\mu_{0}=0,\tau_{0}=1$
5	0.095	0.143
10	0.065	0.104
20	0.044	0.074
50	0.026	0.045
100	0.018	0.031

**Table 3 entropy-23-00190-t003:** Bias in favor of the hypothesis $H_{0}=\left\{0\right\}$ with a $N(\mu_{0},\tau_{0}^{2})$ prior for different sample sizes *n* with $\sigma_{0}=1$ using (Equation 4) (and using (Equation 5) with δ=0.5).

*n*	$\left(\mu_{0},\tau_{0}\right)=\left(1,1\right)$	$\left(\mu_{0},\tau_{0}\right)=\left(0,1\right)$
5	0.323(0.871)	0.451(0.631)
10	0.259(0.747)	0.371(0.516)
20	0.215(0.519)	0.299(0.327)
50	0.153(0.125)	0.219(0.062)
100	0.116(0.006)	0.168(0.002)

**Table 4 entropy-23-00190-t004:** Average bias against $H_{0}=0$ when using a $N(0,\tau_{0}^{2})$ prior for different sample sizes *n*.

*n*	$\tau_{0}=1$	$\tau_{0}=0.5$
5	0.107	0.193
10	0.075	0.146
20	0.051	0.107
50	0.031	0.067
100	0.021	0.046

**Table 5 entropy-23-00190-t005:** Average bias in favor for estimation based on (Equation 11) when using a $N(0,\tau_{0}^{2})$ prior for different sample sizes *n* and difference δ.

*n*	$\left(\mu_{0},\tau_{0}\right)=\left(0,1\right),\delta =1.0$	$\left(\mu_{0},\tau_{0}\right)=\left(0,1\right),\delta =0.5$
5	0.451	0.798
10	0.185	0.690
20	0.025	0.486
50	0.000	0.131
100	0.000	0.009

**Table 6 entropy-23-00190-t006:** Expected half-widths (coverages) of the plausible interval when using a $N(\mu_{0},\tau_{0}^{2})$ prior for different sample sizes *n*.

*n*	$\tau_{0}=1$	$\tau_{0}=0.5$
5	0.625(0.893)	0.491(0.807)
10	0.499(0.925)	0.389(0.854)
20	0.393(0.949)	0.312(0.893)
50	0.281(0.969)	0.231(0.933)
100	0.215(0.979)	0.181(0.954)

**Table 7 entropy-23-00190-t007:** Coverage probabilities for $Pl_{\psi }\left(x\right)$ for the 0.95 quantile in Example 5.

*n*	Frequentist Coverage	Bayesian Coverage
10	0.849	0.896
20	0.895	0.927
50	0.934	0.958
100	0.955	0.973

**Table 8 entropy-23-00190-t008:** Baseline bias against values for prediction for location normal in Example 6.

$\tau_{0}^{2}/\sigma_{0}^{2}$	Bias against $\left(y-\mu_{0}\right)/\sigma_{0}=0$	BIAS against $\left(y-\mu_{0}\right)/\sigma_{0}=1$
1	0.239	0.213
10	0.104	0.100
100	0.031	0.031
1/2	0.270	0.263
1/100	0.316	0.460

## Data Availability

Not applicable.

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
