# Peer review of "Measuring and Controlling Bias for Some Bayesian Inferences and the Relation to Frequentist Criteria"

_entropy, 2021, doi:10.3390/e23020190_

Round 1
Reviewer 1 Report
This manuscript is a nice addition to Dr. Evans’ ongoing development of his theory of statistical evidence. The key concepts are explained clearly, and the approach to bias measurement extends the suite of statistical tools he has been developing as part of his logically cogent approach to the problem.
One issue the authors might want to consider is that the use of rigorous measure theoretic notation, while making the treatment mathematically precise, introduces complexity that will be daunting to the non-specialist. Of course this is not a problem in and of itself: the rigors of one’s specialty should be incorporated into technical work. However, it may be an obstacle to wider appreciation of the approach, e.g., by the philosophical community where, as the authors note, much of the discussion of statistical evidence takes place. That said, most of the manuscript should be accessible even to a non-specialist audience, as the text itself is clear and easy to follow.
I have only some minor editorial suggestions:
(1) At the current (print) font size much of the text in the figures is too small to read (e.g., the y-values in Figures 4, 5, which are important to the argument). It would be good to verify that all figure legends are clearly legible in the final print version.
(2) Should the x-axis label on Figure 1 be mu_* rather than mu?
(3) In figures 4, 5 (and also to some extent 7), the plots are not smooth. I am assuming this is an artifact of numerical issues but a comment would be helpful. Perhaps more importantly, some plots appear to be asymmetric. Is that expected?
(4) On page 24 the text starting at line 825 makes 2 references to Example 1; I believe this should be Example 3.
Reviewer 2 Report
The manuscript titled “Measuring and Controlling Bias for Some Bayesian Inferences and the Relation to Frequentist Criteria “ is of great interest. The theme of Bayesian reasoning is treated with remarkable mastery, highlighting the mathematical and statistical aspects underlying it. As stated by the authors "The major purpose of this paper, …, is to deal with a key criticism of Bayesian methodology, namely, that inferences can be biased because of their dependence on the subjective beliefs of the analyst. This criticism is accepted, but we also assert that this can be dealt with in a logical and scientific fashion as has been demonstrated in this paper".
Given the interest in this topic in the literature, I recommend this publication in the version presented.
Reviewer 3 Report
The paper concerns the measure and control of the bias for some Bayesian inferences and the relation to frequentist criteria. These notions are developed through discussion, examples and theorems. At first glance, the first part of the article is a bit surprising, resembling a journal - scholar's article (with clear examples). But from page 12 things get serious and we understand the big contributions. Overall the document is well written with clear comments and discussions. The numerical work is being taken seriously. Theorems are not trivial and interesting.
I recommend publication of this work subject to the following revision:
o Surprisingly, the term “entropy” is never used in the article. This is a bit surprising since the article is submitted in Entropy. Is it possible to comment in some places on the notion of entropy in the article? Or formulate a response on the adequacy of the conclusions of the article with the scopes of Entropy journal.
o Please avoid "cutting equations at the end of the line to another". This makes reading very difficult in some parts.
o The proofs of Theorems need to be seriously revised in the sense that they are clipped and poorly organized. Please put the main equations in the center (using \ begin {align *} \ end {align *}), improve the presentation and avoid all lines breaks.
Round 2
Reviewer 3 Report
The authors have seriously revised the paper, and I approve the present version. I recommend it for publication.